

# Dynamics of CO₂ and CH₄ fluxes in Red Sea mangrove soils

Jessica Breavington[1,2,3], Alexandra Steckbauer[1,2,3], Chuancheng Fu[1,2,3], Mongi Ennasri[1,2,3], Carlos M. Duarte[1,2,3]

[1]Marine Science Program, Biological and Environmental Science and Engineering Division (BESE), King Abdullah University of Science and Technology (KAUST), Thuwal 23955-6900, Kingdom of Saudi Arabia
[2]Red Sea Research Center (RSRC), King Abdullah University of Science and Technology, King Abdullah University of Science and Technology (KAUST), Thuwal 23955-6900, Kingdom of Saudi Arabia
[3]Computational Bioscience Research Center (CBRC), King Abdullah University of Science and Technology, King Abdullah
University of Science and Technology (KAUST), Thuwal 23955-6900, Kingdom of Saudi Arabia

*Correspondence to*: Jessica Breavington (jessicabreavington@gmail.com)

**Abstract.** Red Sea mangroves have a lower carbon burial rate than the global average, whereby small greenhouse gas fluxes may offset a large proportion of carbon burial. Monthly soil core sampling was conducted across 2 years at two sites within a
central Eastern Red Sea mangrove stand to examine carbon dioxide ($CO_2$) and methane ($CH_4$) fluxes under dry and inundated conditions. Fluxes were highly variable, characterized by a prevalence of low emissions punctuated by bursts of high emissions. At the landward site, average ± SE (median) flux from the soil-air interface was $3111 \pm 929$ (811) µmol $CO_2$ m$^{-2}$ d$^{-1}$ and $1.68 \pm 0.63$ (0.26) µmol $CH_4$ m$^{-2}$ d$^{-1}$ under light conditions, and $8657 \pm 2269$ (1615) µmol $CO_2$ m$^{-2}$ d$^{-1}$ and $0.84 \pm 0.79$ (0.59) µmol $CH_4$ m$^{-2}$ d$^{-1}$ under dark conditions. Average ± SE (median) sea-air fluxes were $-55 \pm 165$ (-79)
µmol $CO_2$ m$^{-2}$ d$^{-1}$ and $0.12 \pm 0.23$ (0.08) µmol $CH_4$ m$^{-2}$ d$^{-1}$ under light conditions, and $27 \pm 48$ (53) µmol $CO_2$ m$^{-2}$ d$^{-1}$ and $0.16 \pm 0.13$ (0.09) µmol $CH_4$ m$^{-2}$ d$^{-1}$ in dark conditions. The seaward site recorded higher $CH_4$ flux, averaging $18.7 \pm 8.18$ (1.7) and $17.1 \pm 4.55$ (7.7) µmol $CH_4$ m$^{-2}$ d$^{-1}$ in light and dark conditions. Mean fluxes offset 94.5 % of carbon burial, with a median of 4.9 % skewed by extreme variability. However, reported $CO_2$ removal by total alkalinity emission from carbonate dissolution greatly exceeded both processes and drives the role of these ecosystems as intense $CO_2$ sinks.





## 1 Introduction

Mangrove forests thrive in estuarine and intertidal zones within latitudes of 0 ° to 40 ° (Rosentreter et al., 2018a), storing a significant amount of organic carbon and providing numerous ecosystem services, including coastal protection and biodiversity enhancement (Curran et al., 2002; Howard et al., 2014). Mangroves offer a promising nature-based solution to mitigate global warming due to their high sequestration of soil organic carbon ($C_{org}$), while offering coastal protection to sea level rise (Duarte et al., 2013). Carbon preservation in mangrove soils is facilitated by waterlogged, anoxic conditions that

limit the decay of organic matter (OM). However, as mangroves exist at the boundary between terrestrial and marine environments, the capacity for carbon sequestration varies depending on multiple factors such as the tidal range, sediment and nutrient inputs.

Mangroves in the Red Sea are subject to extreme environmental conditions that restrict their growth and productivity. The

Red Sea is one of the warmest and most saline seas globally, characterized by oligotrophic and nutrient-limited conditions. Moreover, central Saudi Arabia experiences extreme aridity, with an average annual precipitation of 60 mm (Gabr et al., 2017). Consequently, *Avicennia marina*, the predominant mangrove species in the Red Sea, exists at the thresholds of its physiological tolerance. Due to the absence of permanent rivers, mangroves in the Saudi Arabian Red Sea typically form narrow fringing bands along the coastline. In the central Red Sea, the distribution of mangroves is constrained by the small

tidal range, which is typically less than 1.5 m (Blanco-Sacristán et al., 2022). The oligotrophic conditions prevalent in the Red Sea result in stunted growth and dwarf forms of mangroves due to nutrient limitation (Almahasheer et al., 2016b). As a result, mangroves in the Red Sea have one of the lowest rates carbon sequestration rates, approximately $15 \pm 1$ g $C_{org}$ m$^{-2}$ yr$^{-1}$, compared with a global average estimated at 163 g $C_{org}$ m$^{-2}$ y$^{-1}$ (Almahasheer et al., 2017; Breithaupt et al., 2012; Sanderman et al., 2018).


Greenhouse gas (GHG) fluxes, involving the release of carbon dioxide ($CO_2$) or methane ($CH_4$), in mangrove soils partially offset their role in removing atmospheric $CO_2$, which is at its highest in the past 800,000 years (Tripati et al., 2009), contributing to radiative heating of the atmosphere and a global temperature increase at a rate of 1.7 °C per century since the beginning of the industrial revolution (IPCC, 2014; Marcott et al., 2013). $CH_4$ is the second most important GHG associated

with climate change (Forster et al., 2007), and substantially more potent than $CO_2$, with a global warming potential (GWP) approximately 28 times greater (Myhre et al, 2014). The low carbon sequestration rates of Red Sea mangroves may be offset by GHG fluxes. However, a lack of dynamic estimates for GHG fluxes from arid mangrove soils in the Red Sea preclude such assessment. To date, and to the best of our knowledge, only one other known study has provided estimates of GHG fluxes from mangrove soil over a short period in this region (Sea et al., 2018). Therefore, it is difficult to reliably quantify

the role of GHG emissions in offsetting $CO_2$ removal by carbon sequestration in Red Sea mangrove soils, which is important





to creating accurate carbon budgets for arid mangroves. Furthermore, GHG flux estimations exhibit wide variation due to factors such as location, environmental conditions, and study design.

Intertidal conditions in mangrove forests allow for flux measurements directly from the soil to the air (soil-air interface) or
through the sea-air interface, with different transfer velocity equations introducing variability in the flux estimates (Akhand et al., 2021; Call et al., 2015). Additionally, flux measurements can be measured *in situ* or through controlled *ex situ* laboratory studies, with variations in chamber design, that can be closed, or open with circulating air. Recent advancements in measurement technology, particularly with the growing use of cavity ring-down spectroscopy (CRDS), enable high-accuracy measurements even at low gas concentrations, but accurate comparison with other methods, such as eddy flux
covariance can be challenging (Brannon et al., 2016). Furthermore, environmental variables and physiochemical properties should be considered to comprehensively understand the variability of GHG emissions from mangrove soils. A comprehensive understanding of carbon stores and emissions in mangrove ecosystems is required to accurately determine the net climate benefits from mangrove coverage and restoration efforts (Lovelock et al., 2022). The Red Sea is one of the few regions where mangrove coverage has been steadily increasing over the past four decades, underscoring the importance
of accurate carbon budgets for Red Sea mangroves (Almahasheer et al., 2016a).

Here, we quantify the dynamics of $CO_2$ and $CH_4$ fluxes from mangrove soils in a mangrove stand in the highly arid Central Red Sea to assess the scale of soil carbon burial offset by GHG flux. We also test the effect of various physical and chemical soil properties on GHG fluxes. This study represents the first effort to simultaneously measure $CO_2$ and $CH_4$ fluxes from
both the sea-air and soil-air interfaces in Red Sea mangroves over a time series relevant to providing needed insights into the dynamics of carbon cycling in this unique ecosystem.

## 2 Methods

### 2.1 Sampling location

Sampling was conducted at two adjacent monospecific *Avicenna marina* mangrove stands in Thuwal, on the eastern coast of
the Central Red Sea (22.340787°N, 39.087991°E) (Fig. 1). Soil cores for $CO_2$ and $CH_4$ flux were collected over two years, from April 2021 to May 2023, on a monthly basis, except when this was prevented by logistical challenges. The first sampling location was approximately 150 m inland from the coast, referred to as the landward site, with an elevation approximately 0.75 m above sea level. This landward site was characterised by a strong seasonal microtidal influence, with a tidal range of less than 0.5 m, resulting in a scarce tidal inundation during the summer months and a more regular inundation
during winter. The second site was located approximately 200 m from the landward site, referred to as the seaward site. Sampling of this second site was conducted over a narrower time window between September and October 2022: weekly sampling for two consecutive weeks, followed by a two-week break to minimise disturbance to the site and allow for a



greater range in temperature, and then resumed for another two weeks of weekly sampling, resulting in a total of four sampling events. This seaward site experienced continuous water coverage across the sampling period and resultingly, was subject to fewer environmental extremes than the landward site.


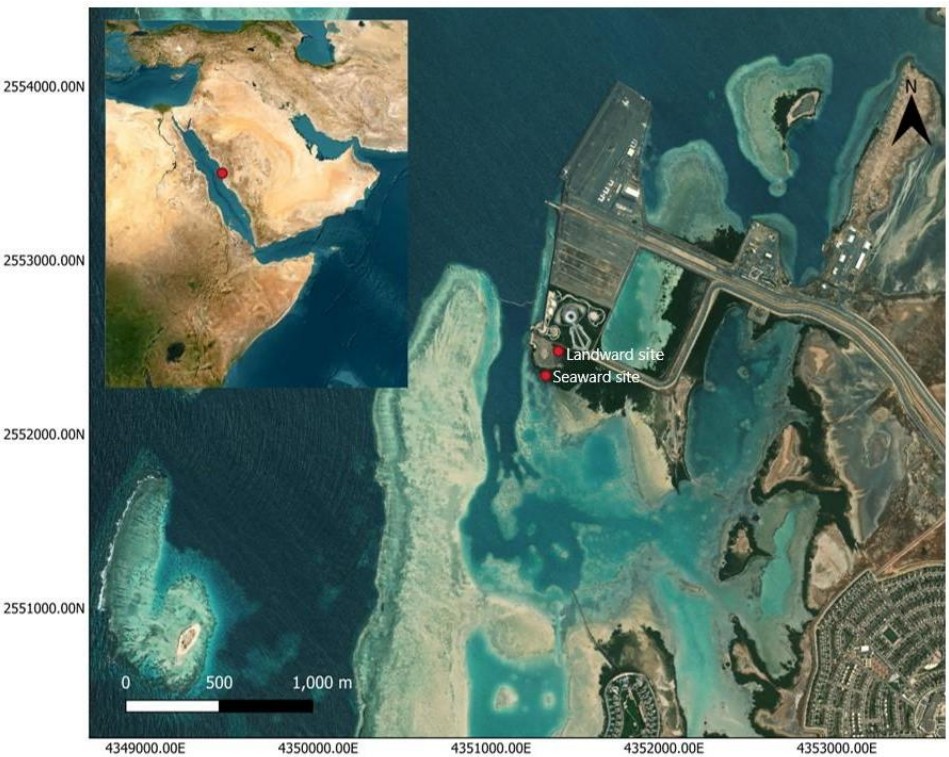

**Figure 1: Mangrove sampling sites as indicated by red circles. Inset: Location of sampling in the Eastern Central Red Sea (from ESRI Satellite).**

## 2.2 Core collection

Two sets of cores were collected each month. The first set of cores comprised four large clear PVC cylinders (height: 30 cm, diameter: 9.6 cm) inserted into the soil to a depth of 10 cm and retrieved without disturbing the soil layers. If water was present during sampling, it was retained within the cylinder up to a maximum depth of 10 cm to ensure a minimum of 10 cm of air for incubation, and without disturbing the soil-water interface. The second set of cores (height: 9 cm, diameter: 2.5 cm) were collected immediately next to the large cores and used to assess the physical and chemical properties of the soil,

including conductivity, total carbon (TC), total organic carbon ($C_{org}$), total nitrogen (TN), bulk density (BD), and water content (WC). Both sets of cores were transferred into a cooler and transported to the lab within an hour after sampling. Temperature and tidal inundation at the site were continuously recorded via *in situ* temperature and water level loggers (U22-001 v2 and U20L-04, Onset Computer Corp., Massachusetts, USA).



### 2.3 Flux measurements

GHG flux was measured from the soil-air interface or the sea-air interface, depending on the presence of water at the time of sampling. The four replicate cores were immediately transported to the laboratory and placed in an incubator (I-30L, Percival, Geneva Scientific LLC, Fontana Wisconsin, USA). Temperature was set to the average temperature in the field as determined by readings from *in situ* temperature loggers (U22-001 v2, Onset Computer Corp., Massachusetts, USA) over the past 72 hours from the time of sampling. After sampling, an airtight lid was fitted to the bottom of the core, and opaque tape

was wrapped around the outside of the core to cover the soil phase, to avoid light exposure to the sides of the soil. The top of each core was left unsealed and kept in the incubator overnight to equilibrate. Immediately before the onset of sampling, the top of the cores were sealed with airtight lids. Three gas samples of 25 mL per core were taken starting at 7 am (T0), after 12 hours of light (T1), and the final sample (T2) after 12 hours of darkness. Gas samples were taken using a syringe and valve system. The syringes with gas samples were connected to a G2201-*i* CRDS analyser (Picarro, Santa Clara, California USA),

coupled to a Small Sample Introduction Module (SSIM2), to measure $CO_2$, $CH_4$, $\delta^{13}C\text{-}CO_2$ and $\delta^{13}C\text{-}CH_4$.

$CO_2$ and $CH_4$ concentrations were converted from dry mole fractions in parts per million (ppm) to $\mu mol\ m^{-2}\ d^{-1}$ (24 hours) using the linear change in concentration between the 12-hour sampling periods (Brannon et al., 2016; Tete et al., 2015) (Eq. 1).

$$F = \frac{dC}{dt}\left(\frac{PV}{RAT}\right) \qquad [\text{Eq. 1}]$$

Where F is flux of $CO_2$ or $CH_4$ ($\mu mol\ m^{-2}$); $dC/dt$ is the linear concentration change of $CO_2$ or $CH_4$ over 12 hours from T0 to T1 to measure light fluxes or T1 to T2 to measure fluxes under the dark condition; P is the pressure (Pa) calculated using Boyles law which was used to correct the pressure in the headspace after taking 25 mL air at each sampling point; V is the volume of gas ($m^3$) in the cylinder headspace; R is the ideal gas constant (8.314 J $mol^{-1}\ K^{-1}$); A is the area of soil ($m^2$); and T is temperature (K).


The $CO_2$ equivalent (g $CO_2$-eq $m^{-2}\ y^{-1}$) of the combined $CO_2$ and $CH_4$ fluxes was calculated for the flux across the sea-air and soil-air interfaces (Eqs. 2 & 3). Mangrove carbon storage was calculated using estimates from previous studies in the Red Sea. Using 55 g $CO_2$-eq $m^{-2}\ y^{-1}$ for the soil carbon burial rate (Almahasheer et al., 2017), and 1266 g $CO_2$-eq $m^{-2}\ yr^{-1}$ for $CO_2$ uptake from total alkalinity (TA) enhancement determined at this site (Saderne et al., 2021).


$$\frac{\mu mol\ m^{-2}d^{-1}}{1,000,000} \times 365 = mol\ m^{-2}\ y^{-1} \qquad [\text{Eq. 2}]$$

$$(mol\ CO_2\ m^{-2}\ y^{-1} \times 44) + (mol\ CH_4\ m^{-2}\ y^{-1} \times 16 \times 28) = g\ CO_2 - eq\ m^{-2}\ y^{-1} \qquad [\text{Eq. 3}]$$

Where $CO_2$ = 44 g $mol^{-1}$, $CH_4$ = 16 g $mol^{-1}$, and $CH_4$ global warming potential (GWP) over a 100-year horizon = 28 (IPCC,

140 2014).





## 2.4 Soil chemical and physical variables

The soil samples from the small cores were fully dried at 60 °C. Samples were ground using an agate pestle and mortar for analysis of total carbon (TC), total organic carbon ($C_{org}$), total nitrogen (TN) and soil conductivity ($EC_{1:5}$). For $C_{org}$, a $10 \pm 1$ mg sample was acidified with 5 µl of 3 mol HCL $L^{-1}$ in silver capsules. Samples were dried for 30 min at 60 °C and

acidification was repeated a minimum of 3 times, or until no bubbles were observed during the addition of HCL to remove all carbonates before being fully dried and wrapped in tin capsules for organic elemental analysis (Flash 2000, ThermoFisher Scientific, Massachusetts, USA). Soil organic carbon ($C_{org}$) and inorganic carbon ($C_{inorg}$) for 0-3 cm soil depth was calculated with the following formulas (Eqs. 4 & 5):

$$C_{org} \left( mg\ C_{org}\ cm^{-3} \right) = Bulk\ Density\ (g\ cm^{-3}) \times \left( \frac{TOC\ (\%)}{100} \right) \times 1000 \qquad [\text{Eq. 4}]$$

$$C_{inorg} \left( mg\ C\ cm^{-3} \right) = Bulk\ Density\ (g\ cm^{-3}) \times \left( \frac{TC\ (\%) - TOC\ (\%)}{100} \right) \times 1000 \quad [\text{Eq. 5}]$$

Conductivity was measured using an electrical conductivity (EC) sensor (Inlab® 738 ISM, Mettler Toledo, Schwerzenbach, Switzerland). Prior to measurement the sensor was calibrated with 12.88 mS/cm potassium chloride as produced by the manufacturer (Mettler Toledo). For the surface soil, $5 \pm 0.01$g of soil was used with 25 mL water for a 1:5 ratio of 1 part soil

to 5 parts Milli-Q water. The samples were mixed on an orbital shaker (VWR©) following a typical protocol for the $EC_{1:5}$ method for high salinity soils (Hardie and Doyle, 2012; Kargas et al., 2018).

## 2.5 Data analysis

Differences in mean soil properties and GHG flux between sampling sites, and wet and dry conditions were evaluated for significance by means of Mann-Whitney U test in R Studio (v.4.1.2). A random forest algorithm was used to model the

influence of environmental, and temporal variables on $CO_2$ flux in light and dark conditions through the use of regression trees utilizing bootstrapping techniques (Breiman, 2001). The models were built in Python v.3.9.13 and Jupyter Notebook v.6.4.12 using the RandomForestRegressor from the SciKit-Learn package v.1.0.2. Only data from the landward site was used in the models due to the greater number of observations and longer sampling period. 80% of data was randomly selected and used for training, with the remaining 20% used for validation.


To optimize model accuracy and minimise overfitting, we utilized the $R^2$ metric, which is an easy-to-interpret standardised measure of linear association (Fox et al., 2017), and implemented a 5-fold cross-validation to assess how the model generalizes to unseen data and reduces the risk of overfitting. Hyperparameter tuning for the number of trees, maximum depth, minimum sample split, and minimum sample leaf, was utilized to maximise the $R^2$ metric. Furthermore, a baseline

accuracy threshold was defined for feature selection, where $R^2 \geq 0.6$ and the average 5-fold cross-validation (CV) score $\geq$ 0.4. Backward elimination of variables based on these performance metrics was systematically performed to maximise the number of variables included within each model without sacrificing model performance to ensure the maximum predictive





power without overfitting (Genuer et al., 2010; Speiser et al., 2019). These models were used to map feature importance of the variables retained from the feature selection stage. All other figures were made with the use of ggplot in R Studio
(v.4.1.2).

## 3 Results

### 3.1 Soil properties

The most pronounced variation in soil characteristics between wet and dry sampling conditions at the landward site was observed in conductivity (EC), averaging 22.6 mS cm$^{-1}$ under dry conditions compared to 9.25 mS cm$^{-1}$ under wet conditions
(Table 1), although all locations were classified under the 'extreme' salinity class (Hardie and Doyle, 2012). EC and WC were the only soil properties to demonstrate significant differences ($p<0.001$) under wet and dry sampling conditions at the landward site. The largest contrast between the two sampling sites was evident in the $C_{inorg}$ concentration, with the seaward site exhibiting a significantly higher ($p<0.001$) mean $C_{inorg}$ ($94.51 \pm 3.37$ mg C cm$^{-3}$) compared to the landward site ($66.64 \pm 1.16$ mg C cm$^{-3}$) under wet and dry sampling conditions. Additionally, the seaward site had a lower $C_{org}$ concentration,
averaging 5.53 mg $C_{org}$ cm$^{-3}$ compared to an average of 9.52 mg $C_{org}$ cm$^{-3}$ at the landward site throughout the entire sampling period. $C_{org}$ was significantly greater ($p<0.001$) at the landward site under dry conditions, averaging 2.43 mg $C_{org}$ cm$^{-3}$ more than the seaward site. Under wet conditions, there was a smaller but still significant difference ($p<0.05$) of 2.29 mg $C_{org}$ cm$^{-3}$ between the landward and seaward sites.

**Table 1: Average soil properties (± SE) for the top 3cm of soil in at the landward site in dry and wet sampling conditions, and at the continually inundated seaward site. C:N = C:N (molar ratio), $C_{org}$= Organic carbon, $C_{inorg}$ = Inorganic carbon, WC = Water content, $EC_{1:5}$ = Electrical conductivity (1:5 soil:water ratio). Mean values for sampling sites and conditions with no common letter are significantly different (Mann-Whitney U test, $p<0.05$).**

| Sample location and condition | C:N | $C_{org}$ (mg $C_{org}$ cm$^{-3}$) | $C_{inorg}$ (mg C cm$^{-3}$) | WC (%) | $EC_{1:5}$ (mS cm$^{-1}$) |
|---|---|---|---|---|---|
| **Landward – Dry** | $12.67 \pm 0.43^a$ | $7.96 \pm 0.24^a$ | $65.38 \pm 1.51^a$ | $40.63 \pm 2.30^a$ | $22.61 \pm 1.71^a$ |
| **Landward – Wet** | $12.44 \pm 0.43^a$ | $7.82 \pm 0.46^a$ | $68.43 \pm 1.79^a$ | $49.57 \pm 1.49^b$ | $9.25 \pm 0.47^b$ |
| **Seaward – Wet** | $11.49 \pm 2.29^b$ | $5.53 \pm 0.95^b$ | $93.51 \pm 3.37^b$ | $34.97 \pm 1.16^a$ | $5.71 \pm 0.23^c$ |

### 3.2 Highly variable $CO_2$ and $CH_4$ fluxes

Between April 2021 to May 2023 twenty months were sampled at the landward site. Nine were under inundated conditions measuring flux from the sea-air interface, and eleven months of sampling was under dry conditions, measuring fluxes from the soil-air interface. Five months could not due sampled due to logistical issues. At the landward site the $CO_2$ flux varied



between -3136 µmol $CO_2$ m$^{-2}$ d$^{-1}$ in the light condition to 37,644 µmol $CO_2$ m$^{-2}$ d$^{-1}$ in the dark condition (Fig. 2). The average

fluxes combined across the soil-air and sea-air interfaces were 1686 ± 546 µmol $CO_2$ m$^{-2}$ d$^{-1}$ under the light conditions and

three times larger, 4774 ± 1337 µmol $CO_2$ m$^{-2}$ d$^{-1}$, under dark conditions (Table. 2). The net daily flux over the full

incubation period combining light and dark fluxes was 3178 ± 806 µmol $CO_2$ m$^{-2}$ d$^{-1}$ (range: -811 to 28,048 µmol $CO_2$ m$^{-2}$ d$^{-1}$). On average, the soil was a net source of $CO_2$ to the atmosphere in all conditions except the light $CO_2$ flux from the sea-air

interface at the landward site (-55 µmol $CO_2$ m$^{-2}$ d$^{-1}$).


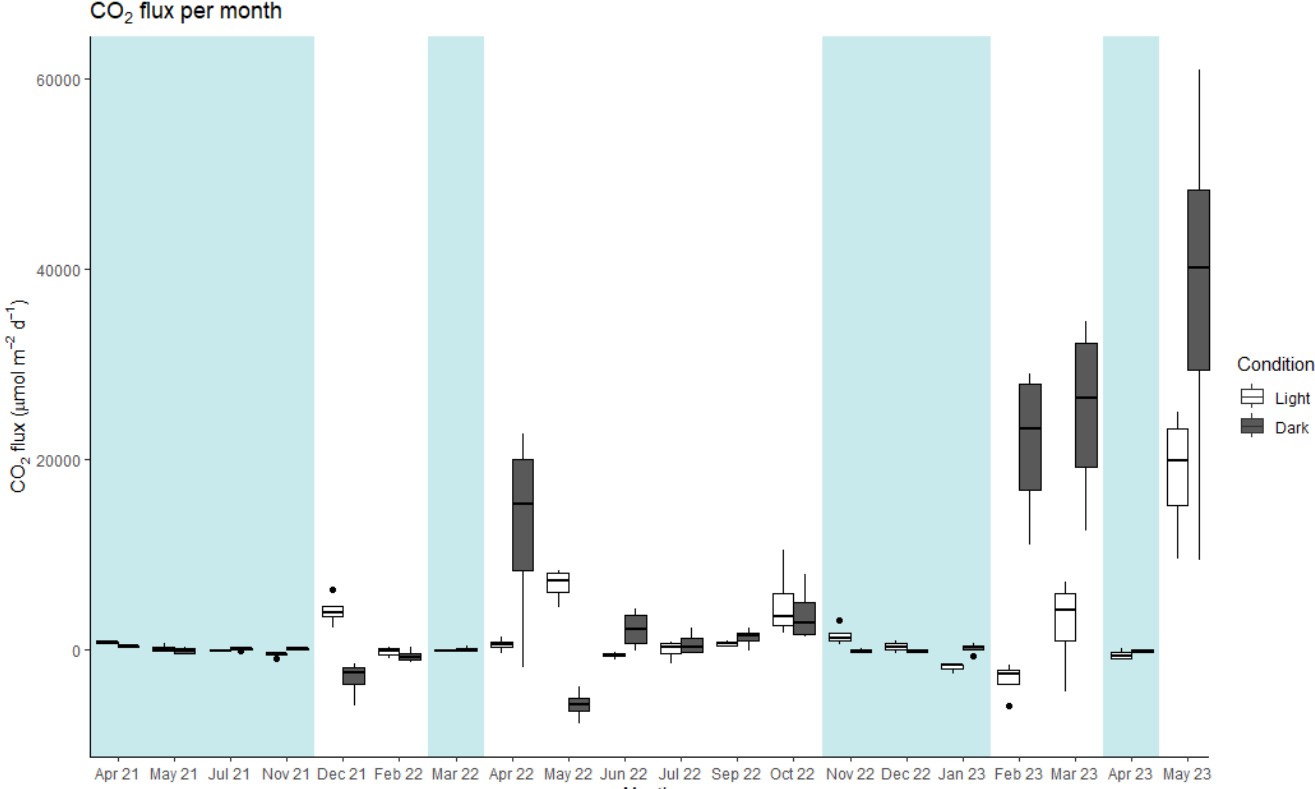

**Figure 2: Median values of CO2 flux for each month and condition (light and dark) at the landward site. The thick line inside the box represents the median value of the data, and 25th and 75th percentiles denoted by the box ends. The whiskers extend to the**

**minimum and maximum values within 1.5 times the interquartile range, outliers are marked by black points. Blue shading: periods of net flux from the sea-air interface. No shading: periods of net flux from the soil air interface. Note: The axis label for the time scale is non-continuous, as months without sampling are not shown.**





The average $CH_4$ flux at the landward site was $0.98 \pm 0.37$ µmol $CH_4$ $m^{-2}$ $d^{-1}$ under the light conditions, $0.54 \pm 0.44$ µmol
$CH_4$ $m^{-2}$ $d^{-1}$ under dark conditions (Fig. 3). The net daily flux over the 24-hour incubation period was $0.74 \pm 0.23$ µmol $CH_4$
$m^{-2}$ $d^{-1}$ (range: -1.47 to 5.71 µmol $CH_4$ $m^{-2}$ $d^{-1}$).

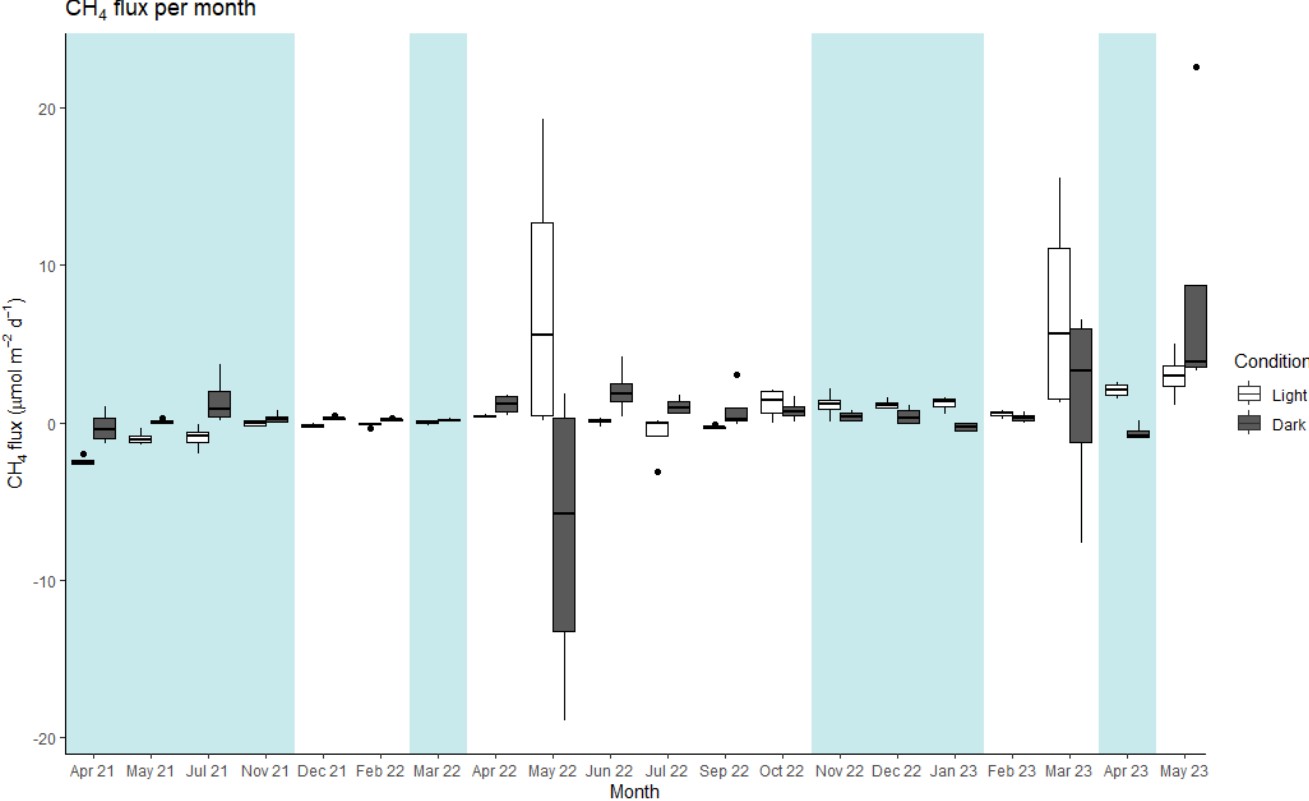

**Figure 3: Median values of $CH_4$ flux for each month and condition (light and dark) at the landward site. The thick line inside the
box represents the median value of the data, and 25th and 75th percentiles denoted by the box ends. The whiskers extend to the
minimum and maximum values within 1.5 times the interquartile range, outliers are marked by black points. Blue shading:
periods of net flux from the sea-air interface. No shading: periods of net flux from the soil air interface. Note: The axis label for the
time scale is non-continuous, as months without sampling are not shown.**

At the seaward site, only sea-air flux was measured given the constant inundation, and there was a lower $CO_2$ flux compared
to the overall mean $CO_2$ flux from landward site (Table 2). However, there was a higher mean and median sea-air $CO_2$ flux
when compared with only the sea-air fluxes from the landward site (Fig. 4). $CH_4$ flux was also significantly higher than that
at the landward site (Table 2). The average flux was 18.67 and 17.12 µmol $CH_4$ $m^{-2}$ $d^{-1}$ in light and dark conditions,
respectively.





The isotopic signature of $CO_2$ averaged -12.02 ± 0.14 ‰ at the landward site and -11.75 ± 0.46 ‰ at the seaward site. Despite the lighter isotope at the landward site there was no significant difference between sites (Mann-Whitney U test, p = 0.0795). The isotopic signature of the $CH_4$ averaged -46.24 ± 0.58 ‰ at the landward site and -48.18 ± 0.67 ‰ at the seaward site, with no significant difference in $\delta^{13}C$-$CH_4$ between sites (Mann-Whitney U test, p = 0.3684).

**Table 2: Summary of $CO_2$ and $CH_4$ fluxes and combined $CO_2$-eq flux offset (using the total flux over the 24-hour incubation) of carbon burial (C burial) and total alkalinity enhancement (TA) for the landward and seaward study sites (Carbon burial data adapted from Almahasheer et al., 2017 and Saderne et al., 2021). Mean values for sampling sites and conditions with no common letter are significantly different (Mann-Whitney U test, p<0.05).**

| Sampling site and condition | Light $CO_2$ ($\mu mol\ CO_2$ $m^{-2}\ d^{-1}$) | Dark $CO_2$ ($\mu mol\ CO_2$ $m^{-2}\ d^{-1}$) | Light $CH_4$ ($\mu mol\ CH_4$ $m^{-2}\ d^{-1}$) | Dark $CH_4$ ($\mu mol\ CH_4$ $m^{-2}\ d^{-1}$) | $CO_2$-eq (g $CO_2$-eq $m^{-2}$ $y^{-1}$) | C burial offset by flux (%) | C burial and TA offset by flux (%) |
|---|---|---|---|---|---|---|---|
| **Landward** | | | | | | | |
| Sea-air mean | -55.2[ac] | 27.5[a] | 0.12[a] | 0.19[a] | -0.2[a] | -0.4[a] | -0.01[a] |
| Soil-air mean | 3110.8[bc] | 8657.4[b] | 1.25[ab] | 0.83[b] | 94.7[b] | 172.1[b] | 7.2[b] |
| Combined mean | 1686.1[c] | 4774.0[ab] | 0.98[a] | 0.54[c] | 52.0[c] | 94.5[c] | 3.9[c] |
| Median | 216.2 | 115.3 | 0.38 | 0.17 | 2.7 | 4.9 | 0.2 |
| Min | -3135.7 | -5799.6 | 0.03 | 0.04 | -71.7 | -130.4 | -5.4 |
| Max | 18547.2 | 37644.0 | 5.71 | 3.96 | 452.0 | 821.8 | 34.2 |
| **Seaward** | | | | | | | |
| Sea-air mean | 2832.5[bc] | 2244.9[b] | 18.67[b] | 17.12[d] | 43.7[b] | 79.5[b] | 3.3[b] |
| Median | 2187.7 | 1669.4 | 1.75 | 7.69 | 31.7 | 57.7 | 2.4 |
| Min | -69.4 | -2770.4 | -0.35 | -0.51 | -22.9 | -41.6 | -1.7 |
| Max | 10361.4 | 7785.7 | 101.9 | 51.2 | 158.2 | 287.7 | 12.0 |

Fluxes were generally a net source of $CO_2$-eq to the atmosphere (Table 2). Using a mean estimate, 95% of soil carbon burial was offset by GHG flux at the landward site. However, the estimates were highly skewed, so that the mean value does not represent the central tendency, which was best represented by the median flux. Median $CO_2$-eq fluxes only offset 4.9 % of the carbon burial rate at the same site. When incorporating the $CO_2$ drawdown of TA enhancement, 3.9 % (mean) and 0.2 % (median) of carbon sequestration potential was offset by the GHG fluxes measured in this study at the landward site (Table

2). At the seaward site, the greater flux and GWP of $CH_4$ resulted in a greater median offset of carbon burial compared to the landward site but the mean offset at the landward site remained higher due to the very large upper-range $CO_2$ fluxes.





Generally, the $CO_2$-eq offset was significantly higher when fluxes were measured between the soil-air interface, compared to measurements between the sea-air interface (Fig. 4).




**Figure 4: Boxplot comparison of mean (red diamond), IQR (boxes), median (black line) and outliers (white circles) of $CO_2$-eq flux**
**across sites, with landward sites separated by dry and wet sampling conditions. "Total" shows the combined g $CO_2$-eq $m^{-2}$ $y^{-1}$ for**
**both $CO_2$ and $CH_4$.**

### 3.3 Drivers of flux variation

Random forest modelling for $CO_2$ flux under light conditions yielded the maximum predictive power, with an $R^2$ value of 0.62, when using only 8 variables. Inclusion of any additional variables resulted in a model performance below the baseline
threshold ($R^2 \geq 0.6$, CV-score $\geq 0.4$). Of the 8 variables, temperature is the most important single variable in correctly




predicting $CO_2$ flux under light conditions with the feature importance of temperature exceeding 0.3, compared to all other variables in the model which have a feature importance below 0.2 (Fig. 5). Temporal variables (year and month) featured among the 8 selected variables, with the year of sampling being the second-most important variable in predicting $CO_2$ flux under light conditions (0.19 importance).

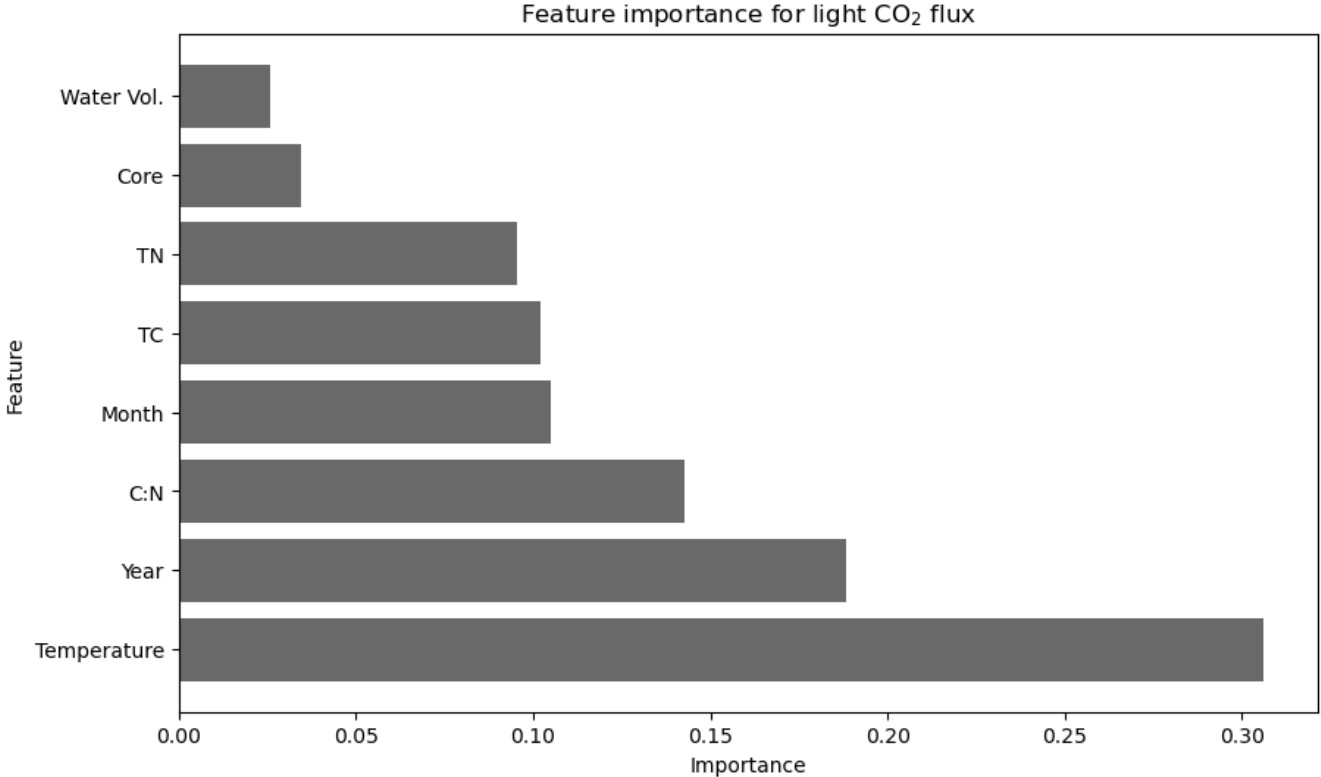


**Figure 5: Importance of environmental, temporal, and physiochemical variables in predicting $CO_2$ flux under light conditions at the landward sampling site ($R^2$=0.62, cross-validation average (n=5) of 0.48 after feature engineering).**

$CO_2$ flux under dark conditions was accurately predicted with the inclusion of 15 out of the 17 possible variables (Fig. 6). The most important single variable in predicting $CO_2$ flux under dark conditions was $\delta^{13}C$-$CH_4$ (0.46 importance) by a large

margin. $\delta^{13}C$-$CH_4$ averaged –47.5 ± 0.25 ‰ in dry conditions and –44.75 ± 1.2 ‰ under wet conditions, with a large range from –54.84 ‰ to –21.12 ‰. There was a negative correlation between $\delta^{13}C$-$CH_4$ and $CO_2$ flux in both dark (-0.2) and light (-0.22) conditions. As with the model for light $CO_2$ flux, the year of sampling was also the second most important feature in predicting dark $CO_2$ flux (0.14 importance). The remaining 13 variables all had a feature importance below 0.1, but this combination contributed towards a high $R^2$ score of 0.63.




In both models the core replicate was of minor importance (Figs. 5 & 6). The season during sampling was not included in the random forest model due to its nature as a categorical variable and high collinearity with other variables. Instead, Kruskal-Wallis ANOVA showed that the season had significant relationships with temperature ($p < 0.001$), water volume in the cores during incubation ($p < 0.05$), dark $CH_4$ flux ($p < 0.05$) and light $CH_4$ flux ($p < 0.05$). However, GHG flux and soil water content (WC) were not significantly influenced by seasonality.


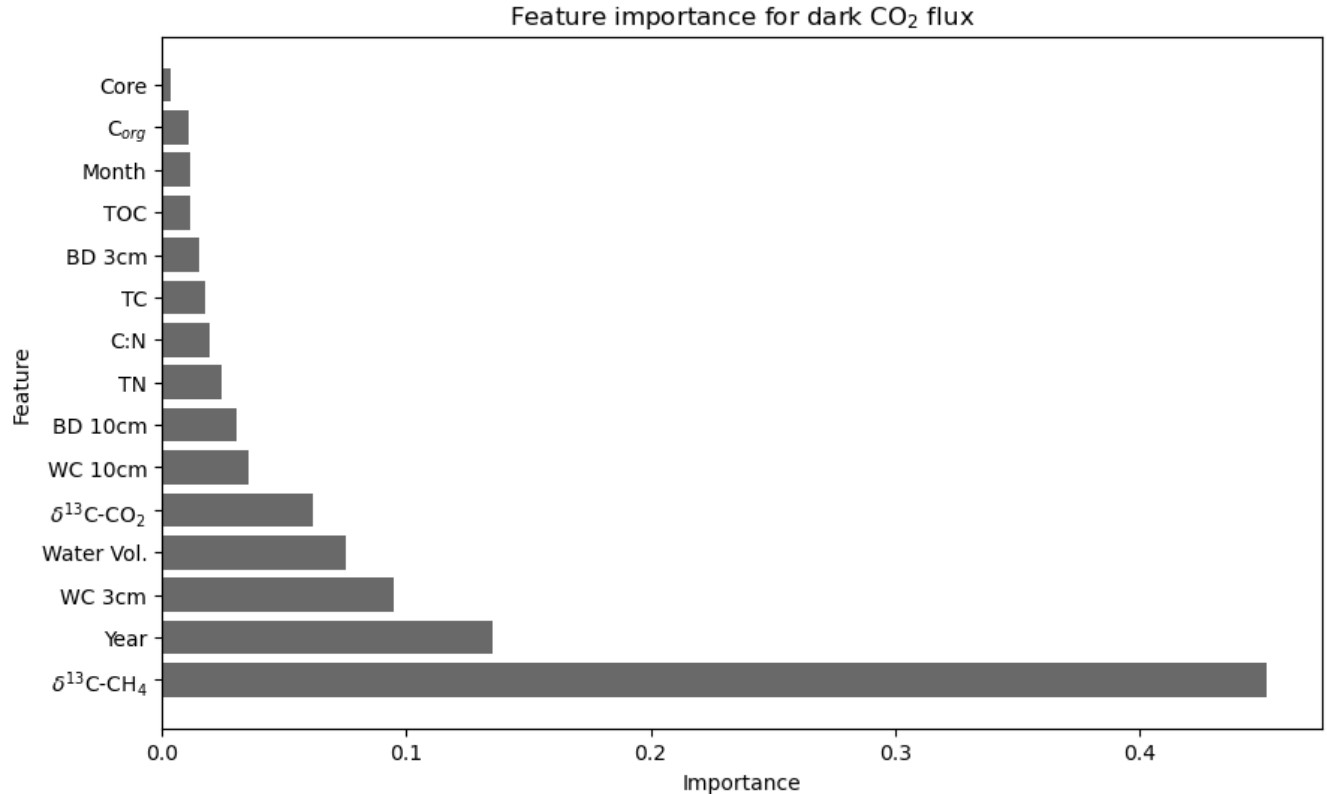

**Figure 6: Importance of environmental, temporal, and physiochemical variables in predicting $CO_2$ flux under dark conditions at the landward sampling site ($R^2$=0.63, cross-validation average (n=5) of 0.43 after feature engineering).**

## 4 Discussion

**4.1 Small but highly variable GHG fluxes**

The $CO_2$ and $CH_4$ fluxes reported in this study are, in general, a small source of GHG emissions, but include episodic events of high flux. The results fall within the lower range of fluxes previously reported for the Red Sea, with $CO_2$ flux between -16,900 to 629,200 µmol $CO_2$ m$^{-2}$ d$^{-1}$, and $CH_4$ flux ranging from -2.1 to 25,974 µmol $CH_4$ m$^{-2}$ d$^{-1}$ (Sea et al., 2018). A review



of 140 mangrove studies reported the global average $CO_2$ flux of 56,800 ± 890 µmol $CO_2$ m$^{-2}$ d$^{-1}$ (Rosentreter et al., 2018b),

while a $CH_4$ flux of 4,557.0 ± 1,102.1 µmol $CH_4$ m$^{-2}$ d$^{-1}$ was found across 54 mangrove studies, with a total of 110 flux observations (Al-Haj and Fulweiler, 2020) (Table. 3). Our results for sea-air fluxes in particular are many orders of magnitude smaller than other studies with similar methodologies (Jacotot et al., 2018). Two defining characteristics of the soil in this study is the low $C_{org}$ and high salinity, which may reduce $CO_2$ and $CH_4$, respectively (Ouyang et al., 2017; Poffenbarger et al., 2011).


While comparisons can, and should, be drawn across different studies, the methodology of the study should be considered when interpreting results. For example, *in situ* studies have the advantage of natural conditions with minimal disturbance caused by sampling, whereas *ex situ* studies, such as incubation techniques, allows for greater control of variables but typically cannot entirely replicate *in situ* conditions such as diel temperature variation, changes in light intensity and

meteorological conditions (Toczydlowski et al., 2020; Sjögersten et al., 2018). For example, one study found mangrove ecosystem flux of CH4 was the most variable on a daily basis due to meteorological variables and plant activities, both of which were excluded in this study (Liu et al., 2022). However, this study utilized incubations to maintain stringent control of environmental variables during the measurement period. The caveat of this approach is that it limits applicability to field conditions, but is useful in separating the effects of individual drivers of GHG flux variation from mangrove soil, and

minimising the number of confounding variables (Bond-Lamberty et al., 2016). An additional element of variation comes from different measurement techniques, as results can differ markedly between laser-based spectrometers, chamber-based systems, and eddy covariance measurements (Brannon et al., 2016; Podgrajsek et al., 2014). All studies compared in table 3 are of *in situ* design, but there are a range of techniques and calculations used. These elements of variability complicate comparison across studies. There is often a large variation in GHG flux across studies and it should be considered whether

this variation is due to environmental conditions or different study designs. For example, in the same study site, $CH_4$ fluxes from eddy covariance measurements have been lower than closed static chamber designs (Gnanamoorthy et al., 2022).

**Table 3: Comparative assessments of average mangrove fluxes under light conditions and standardised to µmol m$^{-2}$ d$^{-1}$ (± SE where data is available), unless otherwise specified. Fluxes for this study calculated using data from both sites. More**

**comprehensive review papers can be found for $CO_2$ (Rosentreter et al., 2018b), and $CH_4$ (Al-Haj and Fulweiler, 2020).**

| Study | Location | $CO_2$ flux (µmol $CO_2$ m$^{-2}$ d$^{-1}$) | $CH_4$ flux (µmol $CH_4$ m$^{-2}$ d$^{-1}$) | Interface |
|---|---|---|---|---|
| This study | Central Red Sea, Saudi Arabia | 5788 ± 1341 | 1.25 ± 0.34 | Soil-air |
| This study | Central Red Sea, Saudi Arabia | 3059 ± 679 | 3.67 ± 1.15 | Sea-air |





| Das and Mandal, 2022 | Sundarbans, India | Range: 17,460 to 70,000 | Range: 100 to 310 | Soil-air |
|---|---|---|---|---|
| Hien et al., 2018 | Northern Viet Nam | 95,500 ± 89,280 | | Soil-air |
| Leopold et al., 2013 | New Caledonia | 91,800 ± 78,200 | | Soil-air |
| Chen et al., 2010 | South China | Range: 560 to 20,560 | Range: 10.1 to 5168.6 | Soil-air |
| Kitpakornsanti et al., 2022 | Thailand | 62,160 ± 22,560 | 92.64 ± 48.24 | Soil-air |
| Rosentreter et al., 2018a | Queensland estuary, Australia | 156,900 ± 94,700 | | Soil-air & sea-air |
| Kitpakornsanti et al., 2022 | Thailand | 39,840 ± 15,840 | 59.28 ± 35.28 | Sea-air |
| Akhand et al., 2021 | Iriomote Island, Japan | 2352 ± 2208 to 54,072 ± 50,976 | | Sea-air |
| Call et al., 2015 | Queensland bay, Australia | Range: 9400 to 629,200 | Range: 13.1 to 632.9 | Sea-air |
| Chen et al., 2010 | South China | Range: 560 to 20,560 | Range: 10.10 to 5168.6 | Soil-air |
| Bouillon et al., 2008 | Global average | 59,000 ± 52,000 | | Sea-air |
| Rosentreter et al., 2018b | Global estimate | 56,800 ± 890 | | N/A |
| Al-Haj and Fulweiler, 2020 | Global estimate | | 4557.0 ± 1,102.1 | N/A |

## 4.2 Drivers of flux variation

Although the landward and seaward study sites were within the same mangrove stand, there were considerably higher fluxes of $CO_2$ and $CH_4$ at the sea-air interface of the seaward site (43.7 g $CO_2$-eq m$^{-2}$ y$^{-1}$), compared to the sea-air interface at the landward site (-0.2 g $CO_2$-eq m$^{-2}$ y$^{-1}$). The main distinguishing environmental factor between the two sites appears to be the frequency and magnitude of tidal inundation as the landward site was microtidal, with long periods without tidal inundation. There is a strong semi-annual seasonal influence on tides in central Red Sea. Extremely hot summer months coincide with low mean sea-level states (Sultan et al., 1996), and in winter, the normally prevailing northwest winds are met by southeast winds, forming the Red Sea Convergence Zone in the centre of the Red Sea, resulting in higher mean sea levels (Langodan et



al., 2017). This is supported by our analysis, showing the significance (p<0.05) of season on the water volume captured in
the soil cores during sampling. There was also a statistically significant seasonal influence on light and dark $CH_4$ flux. This
seasonal effect is likely modulated by temperature variation, which proved to be an important element of light $CO_2$ flux in
our random forest model. Additionally, higher temperatures in summer increase sub-surface soil temperature which can
increase $CH_4$ emissions due to the temperature-dependency of microbial methanogens (Liu et al., 2020). The frequent
absence of tidal inundation in summer exacerbates this effect as the high latent heat capacity of water could otherwise help
regulate soil temperatures. Therefore, seasonality may exert a dual impact on methane emissions, explaining the significance.

A second important factor in $CH_4$ flux is salinity, measured by electrical conductivity in this study which demonstrated a
lower mean in the seaward location. This may explain the higher $CH_4$ emissions from this site as salinity is reported to have a
negative influence on $CH_4$ flux (Liu et al., 2020). Hypersaline mangrove environments are associated with low methane
emissions (Cotovicz et al., 2024; Sea et al., 2018), because high salinity supresses microbial activity and biogeochemical
processes, reducing GHG cycling (Zhu et al., 2021). There is a proposed salinity threshold of 18 ppt, where $CH_4$ flux may
become negligible which is significantly below the salinity found in the Red Sea (Alhassan and Aljahdali, 2021;
Poffenbarger et al., 2011). The causes for the large differences in GHG flux between sites within the same mangrove stand
are not fully resolved. However, it is important to emphasise the need for comprehensive assessments to determine the true
magnitude of GHG flux in a given mangrove ecosystem considering this small-scale variability. Commonly, spatial variation
in GHG fluxes is inferred from a few plots within the study site (Castillo et al., 2017). However, this method is likely to
result in larger errors in estimates without attempting to determine factors driving this variation.

As evidenced by the monthly and site-specific flux variation, environmental and soil physiochemical factors are important in
regulating mangrove soil GHG fluxes. In the literature, there are a multitude of variables suggested to influence $CO_2$ and $CH_4$
flux from mangrove soils.  The variables reported to affect $CO_2$ fluxes include soil organic carbon (SOC), nitrogen,
phosphate, iron, ammonium, porosity, and tidal range (Chen et al., 2010; Jacotot et al., 2018; Sugiana et al., 2023; Wang et
al., 2016). The variables reported to affect $CH_4$ flux include SOC, ammonium, porewater salinity, redox potential, soil
temperature, air temperature and tidal range (Allen et al., 2007; Chen et al., 2010; Jacotot et al., 2018; Sugiana et al., 2023;
Wang et al., 2016). Furthermore, additional factors have been suggested as general influences on overall mangrove GHG
flux such as temperature fluctuations, soil moisture content, soil grain size, and tidal patterns (Hien et al., 2018; Ouyang et
al., 2017). Many of these factors are inferred by a correlational relationship with GHG flux, with many variables likely to be
colinear making causality difficult to determine.

An advantage of the random forest algorithm is that it allows many variables to be taken into account, with the ability to
uncover non-linear relationships, its resistance to outliers, and the ability to test the model on other datasets (Smorkalov,
2022). However, there were variables mentioned above which were found to be important in GHG flux in other studies, but





were not measured in this study, for example ammonium, iron, and soil grain size. There are limitations on the number of variables relative to a fairly small number of observations as in this study (Kiers and Smilde, 2007), along with practical

limitations of time and resources. There is substantial scope in future research to comprehensively investigate more variables than those reported here over a longer sampling period, or with more frequent observations. An analysis of a greater number of chemical and physical characteristics of the soil beyond carbon and nitrogen would be particularly relevant for GHG flux (Nóbrega et al., 2016; Chen et al., 2010). This limitation must be acknowledged when interpreting our results as there may have been significantly important factors which were not measured and thus not considered in our analysis of the most

important drivers of GHG flux.

For the soil, temporal and environmental variables measured in this study, the random forest modelling we conducted suggested temperature to be the most important factor in predicting light $CO_2$ flux and $\delta^{13}C$-$CH_4$ to be the most important factor in accurately predicting dark $CO_2$ flux. To the best of our knowledge, the isotopic signature of methane is a variable

that has not been previously suggested as an important predictor of mangrove $CO_2$ flux. This may be a feature of random forest overfitting or that the $\delta^{13}C$-$CH_4$ signature masks other variables which are the true driving influence of $CO_2$. However, the mean $\delta^{13}C$-$CH_4$ at the landward and seaward sites (-47.2‰ and -48.1‰) were considerably less enriched than the -80.6‰ $\delta^{13}C$-$CH_4$ found in a similar study on Red Sea mangrove GHG flux (Sea et al., 2018). The difference may be due to a number of factors including methanogenesis or oxidisation, although both factors are unlikely to directly influence $CO_2$

emissions. A previous study has found mangroves with the lightest $\delta^{13}C$-$CO_2$ and $\delta^{13}C$-$CH_4$ to have the lowest $CO_2$ flux further suggesting there may be a link between $\delta^{13}C$-$CH_4$ and $CO_2$ flux (Sea et al., 2018). Variations in $\delta^{13}C$-$CH_4$ is highly likely to be driven by microbial processes, for example, methanotrophic bacteria which oxidize a fraction of total $CH_4$ production resulting in a more positive $\delta^{13}C$-$CH_4$. A range of −65 ‰ to −50 ‰, similar to this study, found that aceticlastic methanogenesis (produced from acetate) dominates (Ouyang et al., 2024; Teh et al., 2005). Additionally, in a previous study

on mangrove forests in Mexico it was found that 30–70% of the total $CO_2$ measured was produced by methanogenesis (Sanchez-Carrillo et al., 2021). Anaerobic oxidation can also form $CO_2$ (Shukla et al., 2013). These are possible explanations for our results demonstrating high importance of $\delta^{13}C$-$CH_4$ as a predictor of $CO_2$ flux, however investigation of the soil microbial community would be necessary to draw solid conclusions on the mechanisms underlying this statistical relationship.


From our random forest models, the most important soil variables for $CO_2$ flux were C:N, and TC for light conditions, and soil water content (WC-3cm), and water volume for dark conditions. All factors have previously been documented to play a role in $CO_2$ emissions (Chen et al., 2010). Preservation of TC is related to factors such as water level and inundation time, and where low OC burial efficiency increases soil respiration (Breithaupt et al., 2019). C:N is a good predictor of soil

microbial respiration (Fang and Moncrieff, 2005), and has previously been found to have a significant positive correlation with mangrove $CO_2$ flux (Hien et al., 2018). Furthermore, soil respiration exhibits diurnal patterns which may explain the





high importance of carbon and nitrogen concentrations in predicting light $CO_2$ flux but not dark $CO_2$ flux (Jin et al., 2013). C:N may also be a good predictor for $CO_2$ flux variability because of its relationship with the labile carbon pool, influenced by microbial biomass which will vary by month and season depending on the suitability of conditions for microbial growth

(Padhy et al., 2020). Secondly, soil water content has been found to exert a negative influence on $CO_2$ flux, but have positive relationship with $C_{org}$ (Ouyang et al., 2017). However, there is also likely to be covariation among water content and variables not measured in this study, such as soil porosity, grain size, and density of crab burrows which can increase $CO_2$ flux (Booth et al., 2019; Ouyang et al., 2017). This implies that the interpretation of GHG flux variability should be carefully considered to ensure that non-linear relationships between multiple interrelated variables are accounted for.


In both models, the core was the of minor importance in predicting $CO_2$ flux, which shows good replicability across the four cores sampled each month. The random forest model for the $CO_2$ dark condition had a good $R^2$ score while maintaining the majority of variables (15 of 17), with a comparable performance with models published in previous studies to predict SOC stock (Moreno Muñoz et al., 2024). However, based on the $R^2$ metric, the model for light $CO_2$ flux performed poorly on a

higher number of variables suggesting that many of these variables simply added 'noise' to the predictions, without adding predictive power (Fox et al., 2017). It is likely the models' performance, particularly for light $CO_2$ could be improved with the addition of other unmeasured factors such as clay or sulphur content which were found to be important predictors of soil $CO_2$ flux in sugarcane with random forest modelling (Tavares et al., 2018). The $CH_4$ flux was not modelled due to the importance of microbial activities in $CH_4$ cycling, which would not be accurately captured by the variables measured in this

study (Das et al., 2018; Liu et al., 2020; Yu et al., 2020).

**4.3 Implications for mangrove carbon budget**

Despite the small magnitude of fluxes reported in this study compared to global estimates, they deserve consideration in the net carbon sequestration of Red Sea mangroves given their low carbon burial rate (Almahasheer et al., 2017). The carbon sequestration offset by the $CO_2$-eq of the combined $CO_2$ and $CH_4$ fluxes measured in this study ranged between –130% and

822%. A negative $CO_2$-eq implies net GHG removal from the atmosphere. There was an important difference between the mean and median offset of carbon sequestration by the combined $CO_2$ and $CH_4$ fluxes, which were 94.5% and 4.9%, respectively. The median estimate is less affected by extreme values, and is, therefore, more representative of the central tendency of the offset, while the mean estimate fully captures the large variability in the long-term dataset of this study. Previous studies have also highlighted extreme variability where global mean emissions of $CH_4$ flux were ~16 times higher

than the median estimate (Al-Haj and Fulweiler, 2020). Highly skewed data is appropriate to use only if it accurately reflecting the true distribution of fluxes and not sampling bias (Rosentreter and Williamson, 2020). In this study, averages are likely to be an accurate statistic, given the controls on sampling location and consistent samples times each month over the full study period. This means that whereas the combined $CO_2$ and $CH_4$ fluxes were relatively small compared to reported



mean organic carbon sequestration by the Red Sea mangrove stands studied, these are subject to occasional large emissions
that offset much of the carbon removed.

Prior studies have found GHG emissions to offset between 9.3 to 32.7% of the organic carbon sequestration of mangrove
forests (Chen et al., 2016). A large component of this variability is dependent whether fluxes are measured between the sea-
air or soil-air interface (Table 3). $CO_2$ emissions, which are the biggest contributor to $CO_2$-eq emissions are greatly affected
when being measured between the sea-air or soil-air interface. When $CO_2$ is released from the soil into the water column it
enters the carbonate system and can be converted to bicarbonate or carbonate ions (Zeebe and Wolf-Gladrow, 2001). As a
result, the majority of $CO_2$ emitted from the soil undergoes dissolution in the water column before it is released to the air.
This explains the lower $CO_2$-eq from the sea-air interface of $-0.4$ g $CO_2$-eq m$^{-2}$ y$^{-1}$ compared to 172.1 g $CO_2$-eq m$^{-2}$ y$^{-1}$ for the
soil-air interface when soils are directly exposed to air. Typically, when fluxes are measured from the sea-air interface,
equilibration equations are used to account for the changes in carbonate chemistry in the seawater (Akhand et al., 2021; Call
et al., 2015). However, the aim of this study was to compare GHG flux to the air between interfaces, so the calculations used
here only consider linear changes in concentration across timepoints emphasising diffusive fluxes to the atmosphere over
other methods of gas transfer such as bubble ebullition (Jacotot et al., 2018). Overall, the $CO_2$-eq released to the atmosphere
is a significant offset to carbon burial given the carbon burial rate of Red Sea mangroves is just 55 g $CO_2$-eq m$^{-2}$ y$^{-1}$ (15 g
$C_{org}$ m$^{-2}$ y$^{-1}$), over 10-fold lower than the global average of 598 g $CO_2$-eq m$^{-2}$ y$^{-1}$ (163 g $C_{org}$ m$^{-2}$ y$^{-1}$) (Almahasheer et al.,
2017; Breithaupt et al., 2012).

However, In the Red Sea region mangrove soils have a high carbonate content, our estimates of $C_{inorg}$ fall within the upper
range of previously reported figures for Red Sea mangroves, which are higher than global average estimates (Garcias-Bonet
et al., 2019; Saderne et al., 2019). Furthermore, mangrove soil in the same location as the present study has 76% $\pm$ 2% (dry
weight) $CaCO_3$, which is attributed to their growth on underlying carbonate platforms formed by Pleistocene coral reefs
(Saderne et al., 2018). As a result, there is an additional factor to consider; the role of total alkalinity (TA) enhancement from
carbonate dissolution in the mangrove soils, which increases the capacity for seawater to absorb $CO_2$ from the atmosphere
(Alongi, 2022; Saderne et al., 2019). Mangroves in the Red Sea are characterized as important TA sources (Saderne et al.,
2019), which are driven by high metabolic activity in their soil and multi-stage biogeochemical processes such as carbonate
dissolution, denitrification, sulfate reduction, and ammonification (Baldry et al., 2020; Saderne et al., 2021; Sippo et al.,
2016).

$CaCO_3$ dissolution is particularly relevant to the central Red Sea, as one mole of dissolved $CaCO_3$ results in the uptake of 0.6
mol of atmospheric $CO_2$ (Frankignoulle et al., 1995). The dissolution of the large $CaCO_3$ pool in the soils of Red Sea
mangroves present a substantial additional carbon sink, exceeding by 23-fold the $C_{org}$ burial rate for the central Red Sea
(Almahasheer et al., 2017; Saderne et al., 2021). Although the lower carbon burial in soil means GHG fluxes are a large

offset to the soil carbon burial, TA enhancement brings the carbon sink value of the mangrove stand in this study to 360 g C $m^{-2}$ $yr^{-1}$, which is 2.2-fold above global mean mangrove $C_{org}$ (Saderne et al., 2021). In the present study the $CO_2$-eq of GHG
fluxes represent a small offset (3.9% on average) to the combined carbon sequestration of this mangrove stand when accounting for carbon burial and TA enhancement combined.

## 5 Conclusion

The long-term flux variability captured in this study provides valuable insights into the role of GHG flux in offsetting carbon burial in Red Sea mangrove soils. Our study involved an improved temporal resolution, in terms of the overall duration and
frequency of assessments, beyond most previous assessments. This is important because our results show that $CO_2$ and $CH_4$ fluxes are typically a small carbon offset compared to carbon burial in soils but is punctuated with episodic GHG emission bursts that suffice to offset a large fraction of carbon burial. This aspect of GHG flux dynamics may be missed by studies with poorer temporal coverage.

When considering the carbon budget of the Central Red Sea mangrove stand considered in this study, our results show the
overriding importance of TA enhancement from carbonate dissolution, which is emerging as a major component of mangrove $CO_2$ removal, not yet captured in blue carbon projects. Our results also showed the direct exposure of mangrove soils to the atmosphere drastically enhances GHG emissions compared to emissions during tidal flooding. Environmental conditions helped explain variability in $CO_2$ emissions, whereas those in $CH_4$ emissions seem to be dominated by the dynamics of the microbial community responsible for methanogenesis and methane oxidation.

*Code and data availability*

All data to support the findings of this study are available in FigShare. Raw data for the landward site available at: 10.6084/m9.figshare.26085898. Combined site data across sea-air and soil-air interfaces available at: 10.6084/m9.figshare.26085928. Code and associated data for Random Forest algorithms available at: 10.6084/m9.figshare.26085940.

*Competing interests.*

The authors declare that they have no conflict of interest.

*Author contribution*

C.M.D conceived the research. A.S, C.F, J.B, and C.M.D designed the study. A.S, C.F, and J.B performed the field



sampling. A.S, C.F, J.B, and M.E performed laboratory work. M.E performed gas measurements. A.S, C.F, and J.B conducted data analysis. J.B produced display figures and wrote the manuscript. All authors contributed and approved the manuscript.

*Acknowledgements*

This research was funded by King Abdullah University of Science and Technology through baseline (BAS/1/1071- 01-01) funding for CMD. We extend thanks to Asalata Kotikalapudi for help with sample processing, Reny P. Devassy, Jennifer Thompson, Naira Pluma, Elisa Laiolo and Anastasiia Martynova for their assistance with fieldwork. We thank Larissa Frühe for her assistance with R plots.



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
