# Peer review of "Dynamics of CO2 and CH4 fluxes in Red Sea mangrove soils"

_EGUsphere, 2024_

## Author Comment (AC1)

|  | Comment | Response and action taken |
|---|---|---|
| Reviewer 1 | | |
| Overall comment | This is a very interesting, nicely conducted study that deserves publication. The experiment has been conducted with care and the research question is clear. However, some methods need to be described in more detail, some results should be reported in more detail and some sections of the discussion should be expanded to exploit the scientific benefit that the data harbours. | We would like to thank reviewer 1 for taking the time to give their thorough feedback and useful comments on our paper. We expanded the different sections as requested to explain the methodology and results in more detail, and also expanded on the discussion. For in depth replies, actions taken, and changes in the manuscript please see the comments below. |
| Major comments | General:
Were there no roots in the soil/sediment cores? If mangrove soils store carbon sequestered by the trees, this carbon must get into the soil somehow. | Roots and undecomposed organic matter were avoided where possible as the aim was to estimate greenhouse gas (GHG) emissions from the soil rather than the mangrove trees. Given the thickness of roots at the sampling depth, these were relatively easy to avoid. It is well documented that the density of pneumatophores is related to GHG emissions (e.g. Lin et al, 2021; Sheng et al, 2021). The soil organic carbon comes from a combination of decomposed mangrove roots, leaves, and organic matter from other ecosystems (allochthonous inputs), which is decayed by primary consumers.

Action taken:
To explain better, we added a sentence in the methods section explaining that we avoided the roots when sampling.

It now reads:
*"While sampling the sediment, roots and undecomposed organic matter were avoided as the aim was to estimate GHG emission from the soil rather than the mangrove trees."* |
| | L 40-43: Maybe add a sentence elaborating how this aridity influences mangrove growth. Does it also play a role in selecting the dominant mangrove species? | Thank you for the suggestion. We updated the text to clearly define *Avicennia marina* as the dominant mangrove species in the Red Sea and added more information on the adaptations of *A. marina*.

It now reads:
*"Consequently, Avicennia marina, the dominant mangrove species in the Red Sea, is better adapted to the high salinity and aridity, and found predominantly as monospecific mangrove stands (Khalil, 2015). Rhizhophora mucronata are also found within the Red Sea but predominantly in Southern regions where there is lower salinity (Khalil, 2015; Ahmed & Abdel-Hamid, 2007)."*

Moreover, we added the following text: |

| | | |
|---|---|---|
| | | *"The conditions in the Red Sea result in reduced growth of A. marina, with trees only reaching 2-3 meters compared to over 16 meters in Australia (Mackey, 1993) The high temperatures, and nutrient-limited conditions prevalent in the Red Sea result in stunted growth and dwarf forms of mangroves (Almahasheer et al., 2016)."* |
| Minor comments | L 44: Add "also" following "…mangroves is…" as the small tidal range is probably not the only constraint on mangrove distribution. | Thank you for the correction. We modified the text accordingly.

 It now reads: *"In the central Red Sea, the distribution of mangroves is also constrained by the small tidal range, which is typically less than 1.5 m (Blanco-Sacristán et al., 2022)."* |
| | L 59: What did Sea et al. (2018) find out? It is only one study, but probably one with some results. | In the Sea *et al.* (2018) paper it reads: Diel $CO_2$ and $CH_4$ emission rates ranged from −3452 to 7500 µmol $CO_2$ m$^{-2}$ d$^{-1}$ and from 0.9 to 13.3 µmol $CH_4$ m$^{-2}$ d$^{-1}$ respectively

 It now reads:
 *"Fluxes have been found to range from −3452 to 7500 µmol $CO_2$ m$^{-2}$ d$^{-1}$ and 0.9 to 13.3 µmol $CH_4$ m$^{-2}$ d$^{-1}$ across different locations in the Red Sea (Sea et al., 2018)."* |
| | L 115-116: I have a hard time imagining that 1 2h of equilibration time are sufficient. Did you test whether these 12 h suffice indeed? | This was substandard word choice and we apologize for the confusion. For clarification, the term "Equilibration" was only intended to mean the time between closing the top lid and the T0 measurement, which has now been replaced by the term 'stabilization' (Garcias-Bonet & Duarte, 2017).

 In April, May and July (2021) collection was in the early morning. After transportation to the lab the cores were placed into the incubator to keep them at target temprature. One by one, bottom and top lid were exchanged from "collection lids" to air-tight lids used for the duration of the experiment. After sealing/closing the lids, we allowed 1 hour before taking the T0 measurement (following the established protocol of Sea et al., 2018).

 Due to variability in the tides, we decided to move sampling time of the soil cores to the afternoon on the day before. This was necessary due to unpredictable tidal conditions and logistical challenges of early morning sampling. The cores were then stored in the incubator at target temperature (with open top lid) approximately 12 hours until closing them and taking T0 measurement. Leaving the cores unsealed and undisturbed for this period was to allow for regular gas exchange following disturbance of the soil caused by collection in the field. Moreover, we wanted to avoid creating anoxic conditions in the sediment and water by closing the core too soon before the experiment. In previous studies using similar methodologies (e.g. soil incubation, gas sampling), soil cores were left for a period between 1 hour to 48 hours before the first gas sample collection (Kristensen et al., 2000; Werner et al., 2014; Christiansen et al., 2015; Perez-Villalona et al., 2015; Sea et al., 2018). If there |

| | | was water present, extra water was sampled with the sediment cores and also placed in the incubator to keep the temperature stable. In the morning of the start of the experiment, water was exchanged, if present, 1 hour before the collection of the first gas sample (T0) ( Kristensen et al., 2000; Sea et al., 2018).

The average $CO_2$ concentration (ppm) for T0 across the whole duration of the study was 547 ± 88 ppm, within a normal range for a laboratory during a period of low human activity (Hussin et al., 2017). In April, May and Jul the average was 533ppm, and the T0 measurement was not significantly affected by early morning sample collection, compared to collection the afternoon before the T0 measurement.

Action taken: The text in the methods has been updated to clearly reflect this and minimize ambiguities.

It now reads:
*"Two sets of cores were collected each month. The first set of cores comprised four large clear PVC cylinders (height: 30 cm, diameter: 9.6 cm) inserted into the soil to a depth of 10 cm and retrieved without disturbing the soil layers. If water was present during sampling, it was retained within the cylinder up to a maximum height of 10 cm to ensure a minimum of 10 cm of air headspace for incubation, and without disturbing the soil-water interface. Initially, in April, May, and July (2021) collection of the soil cores was conducted in the early morning hours allowing for sufficient time to transport and process soil cores with a stabilisation period for 1 hour between sealing the core and taking the T0 gas sample at 7am (following the protocol of Sea et al, 2018). Subsequent sampling events, until the study conclusion, were conducted late afternoon on the day before for logistical reasons, with the cores left unsealed in the incubator under darkness to mirror night-time conditions. Leaving the cores unsealed and undisturbed was to allow for regular gas exchange following disturbance of the soil caused by collection in the field, and to avoid the creation of anoxic conditions in the sediment and water overnight before the start of the experiment. If water was present at the time of sample collection, extra water was sampled with the sediment cores and also placed in the incubator to keep the temperature stable. On the morning of the start of the experiment, water was exchanged, and the air-water interface of the sealed cores was allowed to stabilize for 1-hour before the collection of the T0 gas measurement at 7am (following the protocol of Sea et al, 2018). The 1-hour stabilization was not required for cores without the water phase there was no water to exchange, and no water-soil interface to influence gas exchange dynamics. There was no significant difference in T0 concentrations with or without water."* |
| | L 117-118: What is "light"? Laboratory light intensity? Was there a climate chamber to provide the high light intensity prevailing in the region? Or were the cores outside the lab for the "light" phase? | The text was modified to specify the light intensity within the laboratory incubator, which was 125 micromoles m$^{-2}$/sec$^{-1}$ of light irradiance according to the manufacturer |

| | | with the lights set to 100% brightness, reflecting the light intensity in the study region. |
| --- | --- | --- |
| | | It now reads: *"Three gas samples of 25 mL per core were taken starting at 7 am (T0), after 12 hours of light (T1), and the final sample (T2) after 12 hours of darkness. For the duration of the light condition, incubator lights were set to 100 % intensity at 125 μ mol m$^{-2}$/sec$^{-1}$ irradiance (I-30L, Percival, Geneva Scientific LLC, Fontana Wisconsin, USA)"* |
| | L 149-150: How was bulk density determined? | We apologize for the oversight of adding the equation to the main manuscript.

Action taken: We added equation to the main text, which can also be found in supplementary material.

It now reads: *"The soil samples from the small cores were dried at 60 °C to a constant weight. Bulk density was calculated using the equation below (Howard et al., 2014).*

*Soil bulk density (g cm$^{-3}$) = Oven–dry sample mass (g) / Sample volume (m$^3$)"* |
| | L 185-194: It would be nice to see, somewhere in this passage, the %Corg, which is a useful parameter in soil science. | %TOC, with standard error, has been added to this section.

It now reads: *"Additionally, the seaward site had a lower Corg concentration, averaging 5.53 mg C$_{org}$ cm$^{-3}$ (0.34% ± 0.017%) compared to an average of 9.52 mg C$_{org}$ cm$^{-3}$ (0.72% ± 0.021%) at the landward site throughout the entire sampling period."* |
| | L 310-311: Why is this sentence underlined? | Thank you for identifying this error, the issue has been fixed. |
| | L 338-339: Yes, these differences are not fully resolved. But one reason could be different kinds of transport processes inducing CH4 release (ebullitions vas. Diffusive flux). I suggest elaborating this issue a little. | We agree with the comment. This study did not differentiate between diffusive flux and ebullition, but it is true that variable transport processes are an explanation for these differences.

Action taken: A few sentences on the kinds of transport processes have been added to the discussion.

It now reads: *"The causes for the large differences in GHG flux between sites within the same mangrove stand are not fully resolved, although it is likely that there is microscale variation due, in part, to different gas transport processes. The release of CH$_4$ from the soil via ebullition has particularly high spatial variability within sampling sites* |

| | | *(Baulch, 2011; Chuang et al., 2017). Furthermore, the episodic nature of ebullition events may distort the flux calculation which assumes a linear concentration change over time, as is the case with diffusive flux (Jacotot et al., 2018). The possibility of active ebullition in saline, undisturbed mangrove ecosystems requires further investigation, as to-date, no study has found ebullition to be a significant pathway of $CH_4$ release under these conditions (Cotovicz et al., 2024). Considering this small-scale variability, it is important to emphasise the need for comprehensive assessments in individual mangrove ecosystems as GHG flux is highly site-specific."* |
|---|---|---|
| | L 366-384: This issue should be better exploited, also in the results section. Which soil parameters influence acetoclastic and hydrogenotrophic methanogenesis? When do mangroves grow most strongly in the Red sea? This should be the time of a higher rate of acetoclastic CH4 production. Does d13C efflux change with season? | To clarify, the measurement of acetoclastic and hydrogenotrophic methanogenesis was not incorporated within our study design, so unfortunately we cannot present this within our results as this would have required additional measurements. E.g. Inhibition of methanogenesis with 2-bromo-ethane sulphonate, measurement of [14]C isotopes (Kotsyurbenko et al., 2004), $\delta^{13}C$ of the methyl carbon of acetate (ac-methyl) (Penning et al., 2006), or radiotracer incubations (Weston et al., 2014).

We have, however, elaborated on the results of the [13]C isotopic signatures by including some additional statistical analysis. The carbon isotope composition of $CO_2$ and $CH_4$ did not significantly change between seasons (Kruskal-Wallis test: $\delta^{13}C$-$CO_2$, p = 0.62; $\delta^{13}C$-$CH_4$, p = 0.66). There is also no specific season of growth for *A. marina* in the Red Sea. There is considerable interannual and geographic variability, which is largely dependent on temperature and humidity (Almahasheer et al., 2016). As there is no defined growth period, unfortunately, isotopic signatures and fluxes cannot be analysed in conjunction with this.

However, to fully address this comment, we have included a Spearman correlation matrix (inserted below) including significance values for all soil parameters, [13]C signatures, fluxes, and environmental conditions (uploaded as supplimentary material). The design of this correlation matrix has been included in the 'Data analysis' section of the methods.

From this analysis, there were several soil and environmental variables significantly correlated with stable isotope signatures. These variables were; electrical conductivity, core replicate, and inorganic carbon for the top 3cm of soil. Another notable finding of this analysis showed there was no significant statistical relationship using the Spearman correlation coefficient between $\delta^{13}C$-$CH_4$ and dark $CO_2$ flux (inserted below), contrary to the random forest model, which suggested it had the highest predictive power. Our interpretation of this, is that there is a complex relationship between $\delta^{13}C$-$CH_4$ and dark $CO_2$ flux but agree with the reviewer on the importance of methanogenesis and have added to the discussion of this. |

| | | | Action taken: Statistical analysis of the $\delta^{13}C$ signature and season has been added to the results, along with the variables significantly correlated with $\delta^{13}C$. The correlation matrix has been added as supplementary material.

It now reads:
"*The $\delta^{13}C$ signature of $CH_4$ and $CO_2$ did not change significantly across seasons. However, significant correlations (p > 0.05) were observed between core replicates and inorganic carbon ($C_{inorg}$-3cm) with $\delta^{13}C$-$CO_2$, and well as between electrical conductivity ($EC_{1:5}$) and $\delta^{13}C$-$CH_4$.*"

Moreover, we have added the following text to the discussion to reflect the non-significant relationship found in the correlation matrix analysis. Further, a few sentences have been added to acknowledge the importance of acetoclastic and hydrogenotrophic methanogenesis and the potential for further study to improve understanding of the $CH_4$ flux from mangrove soils.

It now reads:
"*Notably, there was no statistically significant correlation between $\delta^{13}C$-$CH_4$ and dark $CO_2$ flux, contrary to the random forest model, which suggested $\delta^{13}C$-$CH_4$ had the highest predictive power for dark $CO_2$ flux. This finding may be a result of overfitting from the random forest modeling or there may be more complex non-linear relationships uncovered by machine learning which are not detected by simple correlation.*"

Moreover, we have added the following subsequent text:
"*To better understand the origin and fate of CH4 from mangrove soils, methanogenesis should be studied directly through the determination of $\delta13C$ of the methyl group of acetate (Goevert et al., 2009) or isotope mass-balance approach Sánchez-Carrillo et al., 2021).*" |
| | Figs 2, 3: I have the impression that most high CO2 and CH4 fluxes occur between March and May, in both years. Is this mirrored by the d13C signature of the fluxes? Is this the season of highest plant growth? | | There is no consistent month for highest plant growth, although there is generally peak flowering and propagule development in November and January (Almahasheer et al., 2016). The highest growth for *A. marina* between March and May is unlikely as this is when temperatures are increasing, there is little to no rain, and tidal inundation becomes less frequent. It is more likely GHG flux correlates with environmental conditions favoring enhanced microbial metabolism. For example, from the correlation matrix, light $CH_4$ flux significantly correlated with temperature (p=0.007) and water content (p=0.009). Dark $CH_4$ flux correlated with water content (p=0.043) and electrical conductivity (p=0.018). $CO_2$ correlated with water volume (light conditions, p=0.008; dark conditions, p=0.032). The $^{13}C$ signature of the fluxes do not have a significant relationship with the magnitude of the flux, except between light $CH_4$ flux and $\delta^{13}C$-$CH_4$ (p=0.005) |

| | | Action taken: The significant correlations with GHG fluxes have been added to the results section 'Drivers of flux variation'.

It now reads:
*"There were several significant correlations relating to environmental and soil properties with GHG flux. Light $CH_4$ flux significantly correlated with temperature (p=0.007) and water content (p=0.009), dark $CH_4$ flux correlated with water content (p=0.043) and electrical conductivity (p=0.018), and $CO_2$ correlated with water volume (light conditions, p=0.008; dark conditions, p=0.032)."*

Additionally, these results have been used to strengthen the 'Drivers of flux variation' section of the discussion, blending this with the discussion on the random forest analysis. |
|---|---|---|
| | Figs 5, 6: The predictive power of the year of sampling is interesting. This should be discussed some more. | The growth and flowering cycles of *A. marina* mangroves in the Red Sea are not annual (Almahasheer et al., 2016). Potentially, the growth cycle of the studied mangrove stand may have changed across the multi-year duration of the study, thus giving the year a high predictive power in the random forest modeling, but this cannot be verified. Additionally, water was present during 4 of the 5 months sampled in 2021, whereas other years were dominated by dry sampling conditions, this may be artificially inflating the predictive power of the year. There were also climatic variables and extreme weather patterns for the region, particularly in 2023 which may explain the predictive power of the year (Van Dijk et al., 2023).

It now reads:
*"In both light and dark models, the year was the second most important predictor for $CO_2$ flux. The growth and flowering cycles of A. marina mangroves in the Red Sea are not annual (Almahasheer et al., 2016). In theory, increased growth over a given year may result in increased soil carbon pools for microbial respiration, directly impacting GHG flux. However, this cannot be tested as mangrove growth was not measured in the present study. Alternatively, the importance of the year of sampling may be artificially inflated in our models due to the presence of water during 4 of the 5 months sampled in 2021 while subsequent years were dominated by dry sampling conditions. However, there were also climatic variables and extreme weather patterns for the region across the 3-year period. 2023 had widespread greening due to higher-than-average rainfall (Van Dijk et al., 2023), potentially also facilitating mangrove growth. It is likely that a combination of these 3 factors explain the predictive importance of the sampling year, and emphasise the importance of long-term flux measurements to capture variations resulting from climatic changes, and perennial life-cycles."* |

Correlation Matrix with P-Value using Function and Spearman

**References:**

Ahmed, E. A., & Abdel-Hamid, K. A. (2007). Zonation pattern of Avicennia marina and Rhizophora mucronata along the Red Sea Coast, Egypt. World Applied Sciences Journal, 2(4), 283-288.

Almahasheer, H., Duarte, C. M., & Irigoien, X. (2016). Phenology and Growth dynamics of Avicennia marina in the Central Red Sea. Scientific reports, 6(1), 37785.

Baulch, H. M., Dillon, P. J., Maranger, R., & Schiff, S. L. (2011). Diffusive and ebullitive transport of methane and nitrous oxide from streams: Are bubble-mediated fluxes important?. *Journal of Geophysical Research: Biogeosciences*, *116*(G4).

Chuang, P. C., Young, M. B., Dale, A. W., Miller, L. G., Herrera-Silveira, J. A., & Paytan, A. (2017). Methane fluxes from tropical coastal lagoons surrounded by mangroves, Yucatán, Mexico. *Journal of Geophysical Research: Biogeosciences*, *122*(5), 1156-1174.

Cotovicz Jr, L. C., Abril, G., Sanders, C. J., Tait, D. R., Maher, D. T., Sippo, J. Z., ... & Santos, I. R. (2024). Methane oxidation minimizes emissions and offsets to carbon burial in mangroves. *Nature Climate Change*, *14*(3), 275-281.

Christiansen, J. R., Outhwaite, J., & Smukler, S. M. (2015). Comparison of CO2, CH4 and N2O soil-atmosphere exchange measured in static chambers with cavity ring-down spectroscopy and gas chromatography. *Agricultural and Forest Meteorology*, 211, 48-57.

Goevert, D., & Conrad, R. (2009). Effect of substrate concentration on carbon isotope fractionation during acetoclastic methanogenesis by Methanosarcina barkeri and M. acetivorans and in rice field soil. *Applied and Environmental Microbiology,* 75(9), 2605-2612.

Howard, J., Hoyt, S., Isensee, K., Telszewski, M., Pidgeon, E. (eds.) (2014). Coastal Blue Carbon: Methods for assessing carbon stocks and emissions factors in mangroves, tidal salt marshes, and seagrasses. Conservation International, Intergovernmental Oceanographic Commission of UNESCO, International Union for Conservation of Nature. Arlington, Virginia, USA.

Hussin, M., Ismail, M. R., & Ahmad, M. S. (2017). Air-conditioned university laboratories: Comparing CO2 measurement for centralized and split-unit systems. *Journal of King Saud University-Engineering Sciences*, 29(2), 191-201.

Khalil, A. S. (2015). Mangroves of the red sea. *The Red Sea: The formation, morphology, oceanography and environment of a young ocean basin*, 585-597.

Kotsyurbenko, O. R., Chin, K. J., Glagolev, M. V., Stubner, S., Simankova, M. V., Nozhevnikova, A. N., & Conrad, R. (2004). Acetoclastic and hydrogenotrophic methane production and methanogenic populations in an acidic West-Siberian peat bog. *Environmental microbiology*, 6(11), 1159-1173.

Kristensen, E., Andersen, F. Ø., Holmboe, N., Holmer, M., & Thongtham, N. (2000). Carbon and nitrogen mineralization in sediments of the Bangrong mangrove area, Phuket, Thailand. *Aquatic Microbial Ecology*, 22(2), 199-213.

Li, H., Dai, S., Ouyang, Z., Xie, X., Guo, H., Gu, C., ... & Zhao, B. (2018). Multi-scale temporal variation of methane flux and its controls in a subtropical tidal salt marsh in eastern China. *Biogeochemistry*, 137, 163-179.

Lin, C. W., Kao, Y. C., Lin, W. J., Ho, C. W., & Lin, H. J. (2021). Effects of pneumatophore density on methane emissions in mangroves. Forests, 12(3), 314.

Mackey, A. Biomass of the mangrove Avicennia marina (Forsk.) Vierh. near Brisbane, south-eastern Queensland. *Marine and Freshwater Research* 44, 721–725 (1993).

Penning, H., Claus, P., Casper, P., & Conrad, R. (2006). Carbon isotope fractionation during acetoclastic methanogenesis by Methanosaeta concilii in culture and a lake sediment. *Applied and Environmental Microbiology*, 72(8), 5648-5652.

Pérez-Villalona, H., Cornwell, J. C., Ortiz-Zayas, J. R., & Cuevas, E. (2015). Sediment denitrification and nutrient fluxes in the San José Lagoon, a tropical lagoon in the highly urbanized San Juan Bay Estuary, Puerto Rico. *Estuaries and Coasts*, 38, 2259-2278.

Sánchez-Carrillo, S., Garatuza-Payan, J., Sánchez-Andrés, R., Cervantes, F. J., Bartolomé, M. C., Merino-Ibarra, M., & Thalasso, F. (2021). Methane production and oxidation in mangrove soils assessed by stable isotope mass balances. *Water,* 13(13), 1867.

Sheng, N., Wu, F., Liao, B., & Xin, K. (2021). Methane and carbon dioxide emissions from cultivated and native mangrove species in Dongzhai Harbor, Hainan. *Ecological Engineering*, 168, 106285.

Van Dijk, A.I.J.M., H.E. Beck, E. Boergens, R.A.M. de Jeu, W.A. Dorigo, T. Frederikse, A. Güntner, J. Haas, J. Hou, W. Preimesberger, J Rahman, P.R. Rozas Larraondo, R. van der Schalie (2024) Global Water Monitor 2023, Summary Report. Global Water Monitor (www.globalwater.online)

Weston, N. B., Neubauer, S. C., Velinsky, D. J., & Vile, M. A. (2014). Net ecosystem carbon exchange and the greenhouse gas balance of tidal marshes along an estuarine salinity gradient. *Biogeochemistry*, 120, 163-189.

Werner, C., Reiser, K., Dannenmann, M., Hutley, L. B., Jacobeit, J., & Butterbach-Bahl, K. (2014). N 2 O, NO, N 2 and CO 2 emissions from tropical savanna and grassland of northern Australia: an incubation experiment with intact soil cores. *Biogeosciences,* 11(21), 6047-6065.

---

## Author Comment (AC2)

| | Comment | Response and action taken |
|---|---|---|
| **Reviewer 2** | | |
| Overall comment | This manuscript investigates CO2 and CH4 fluxes in arid mangroves along the Red Sea. The main findings are GHG fluxes offsets 95% of soil carbon burial in seaward mangrove sites and become net sources during high emission events. However, when total alkalinity enhancement is incorporated, < 4% of carbon sequestration potential is offset by the GHG fluxes. The study also finds that temperature is the most important single variable in predicting CO2 flux under light conditions, second only to the year of sampling due to temporal interannual variability. This study also looked at the relationship between isotopic signature and found a negative correlation between δ13C-CH4, and CO2 flux in both dark and light conditions, which offer insights into how microbial processing are affecting resulting GHG fluxes. Overall, It's a very well written and novel piece of research. | Thank you for taking the time to review our manuscript and providing valuable comments. We appreciate the positive feedback and have addressed the specific points below. |
| Major comments | The main recommendation I have is for authors to include a couple of sentences to acknowledge limitations related to how incubation technique used in this study could have affected GHG gases relative to the field-based observation such as static chambers and continuous eddy covariance. | We agree that there are limitations to the incubation technique, we have added a short discussion on this and justified the reasons for choosing this method. Namely, better ability to control and manipulate conditions, e.g. constant temperature, and consistent light intensity. Although this is an area that could be extensively discussed, we have tried to keep it brief.

Action taken: Added a paragraph discussing the limitations of incubation studies and issues with comparison across different methods (e.g. *in situ, ex situ*)

It now reads:
*"While comparisons can, and should, be drawn across different studies, the methodology of the study should be considered when interpreting results. For example, in-situ studies have the advantage of natural conditions with minimal disturbance caused by sampling, whereas ex-situ studies, such as incubation techniques, allow for greater control of variables but typically cannot entirely replicate in situ conditions such as diel temperature variation, changes in light intensity and meteorological conditions (Toczydlowski et al., 2020; Sjögersten et al., 2018). For example, one study found mangrove ecosystem flux of CH4 was the most variable on a daily basis due to meteorological variables and plant activities, both of which were* |

| | | |
|---|---|---|
| | | *excluded in this study (Liu et al., 2022). However, this study utilized incubations to maintain stringent control of environmental variables during the measurement period. The caveat of this approach is that it limits applicability to field conditions, but is useful in separating the effects of individual drivers of GHG flux variation from mangrove soil and minimising the number of confounding variables (Bond-Lamberty et al., 2016). An additional element of variation comes from different measurement techniques, as results can differ markedly between laser-based spectrometers, chamber-based systems, and eddy covariance measurements (Brannon et al., 2016; Podgrajsek et al., 2014). All studies compared in Table 3 are of in situ design, but there are a range of techniques and calculations used. These elements of variability complicate comparison across studies. There is often a large variation in GHG flux across studies and it should be considered whether this variation is due to environmental conditions or different study designs. For example, in the same study site, CH4 fluxes from eddy covariance measurements have been lower than closed static chamber designs (Gnanamoorthy et al., 2022)."* |
| | Tables 1 and 2. Consider adding significance test results to this table, e.g., compact letter display. | This is a very useful suggestion, which neatly adds a substantial additional information to our results.

Action taken: The methods have been updated to include the significance tests conducted for the table. CLD has been added to Tables 1 and 2 |
| | Figure 4. I'd remove the left panel. I didn't find this zoomed in graph helpful to visualize and understand your results. | This panel was intended to show the differences in the range and median fluxes between the sea-air interface from the landward and seaward site, which is otherwise obscured by the much larger range of fluxes from the soil-air interface from the landward site, although we can see how this may be visually misleading.

Action taken: The zoomed in element of Figure 4 has been removed and the single figure has been enlarged to fit the page, making the differences in flux easier to see. |
| | I agree with you. No one study will ever account for all possible drivers of GHG fluxes. And you are right, lots of these variables can be autocorrelated or have multicollinearity issues. But the relative importance you | Agreed, our random forest models can only include the variables measured so the results cannot not be taken as absolute importance. The text has been updated to more clearly |

| | | |
|---|---|---|
| | found could have been very different had you included, say, for example, ammonium or Fe2 in your analyses, right? With that in mind, I think you could offer a sentence or two on this limitation and implications for follow up studies. | reflect this.

Action taken: Added to the discussion regarding the limited number of soil and environmental properties included in the study and scope for further research on this. We clarified that the random forest models only considered the variables we chose to measure, and are not representative of all soil, temporal and environmental properties.

It now reads:
*"However, there were variables mentioned above that were found to be important in GHG flux in other studies but were not measured in this study, for example, ammonium, iron, and soil grain size. There are limitations on the number of variables relative to a fairly small number of observations as in this study (Kiers and Smilde, 2007), along with practical limitations of time and resources. There is substantial scope in future research to comprehensively investigate more variables than those reported here over a longer sampling period, or with more frequent observations. An analysis of a greater number of chemical and physical characteristics of the soil beyond carbon and nitrogen would be particularly relevant for GHG flux (Nóbrega et al., 2016; Chen et al., 2010). This limitation must be acknowledged when interpreting our results as there may have been significantly important factors which were not measured and thus not considered in our analysis of the most important drivers of GHG flux."* |
| Minor comments | Ln 43. 'physiological' or 'ecophysiological' instead? | Action taken: Replaced physiochemical with physiological

It now reads:
*"Consequently, Avicennia marina, the predominant mangrove species in the Red Sea, exists at the thresholds of its physiological tolerance."* |
| | Ln 104. 'cores' instead of 'scores' | Thank you for identifying this error. The text has been updated. |
| | Ln 268. Remove 'good' or replace it by 'high'. | We agree with this suggestion and have removed 'good'.

It now reads:
*"Although the remaining 13 variables all had a feature* |

| | | |
|---|---|---|
| | | *importance below 0.1 this combination contributed towards an R score of 0.63.* |
| | Ln 313. below 'the' salinity or below 'salinities' | We have changed 'below salinity' to 'below the salinity'

It now reads:
*"There is a proposed salinity threshold of 18 ppt, where CH₄ flux may become negligible which is significantly below the salinity found in the Red Sea."* |
| | Ln 317. Remove the first 'is' from 'this is method is' | Thank you for pointing this out. The correction has been made.

It now reads:
*"However, this method is likely to result in larger errors in estimates without attempting to determine factors driving this variation."* |
| | Ln 317. 'plotsseaaaaaaaaaaaaa'? | Thank you for identifying this error. The correction has been made |
| | Ln 320. 'physico-chemical' instead? | Indeed, 'physiochemical' should read as 'physicochemical'.

Action taken: physiochemical has been replaced by physicochemical for all occurrences within the manuscript. |

**References:**

Bond-Lamberty, B., Smith, A. P., and Bailey, V.: Temperature and moisture effects on greenhouse gas emissions from deep active-layer boreal soils. *Biogeosciences*, 13(24), 6669-6681, https://doi.org/10.5194/bg-13-6669-2016, 2016.

Gnanamoorthy, P., Chakraborty, S., Nagarajan, R., Ramasubramanian, R., Selvam, V., Burman, P. K. D., ... and Zhang, Y.: Seasonal variation of methane fluxes in a mangrove ecosystem in south India: An eddy covariance-based approach. *Estuaries and Coasts,* 1-16, https://doi.org/10.1007/s12237-021-00988-1, 2022.

Kiers, H. A., & Smilde, A. K.: A comparison of various methods for multivariate regression with highly collinear variables. *Statistical Methods and Applications,* 16, 193- 228, https://doi.org/10.0017/s10260-006-0025-5, 2007.

Liu, J., Valach, A., Baldocchi, D., & Lai, D. Y.: Biophysical controls of ecosystem-scale methane fluxes from a subtropical estuarine mangrove: Multiscale, nonlinearity, asynchrony and causality. *Global Biogeochemical Cycles,* 36(6), e2021GB007179, https://agupubs.onlinelibrary.wiley.com/doi/full/10.1029/2021GB007179, 2022.

Nóbrega, G. N., Ferreira, T. O., Neto, M. S., Queiroz, H. M., Artur, A. G., Mendonça, E. D. S., ... & Otero, X. L. (2016). Edaphic factors controlling summer (rainy season) greenhouse gas emissions (CO2 and CH4) from semiarid mangrove soils (NE-Brazil). *Science of the Total Environment*, 542, 685-693, https://doi.org/10.1016/j.scitotenv.2015.10.108, 2014.

Podgrajsek, E., Sahlée, E., Bastviken, D., Ho lst, J., Lindroth, A., Tranvik, L., et al.: Comparison of floating chamber and eddy covariance measurements of lake greenhouse gas fluxes. *Biogeosciences* 11, 4225 –4233. https://doi.org/10.5194/bg-11-4225-2014, 2014.

Sjögersten, S., Aplin, P., Gauci, V., Peacock, M., Siegenthaler, A., and Turner, B. L.: Temperature response of ex –situ greenhouse gas emissions from tropical peatlands: Interactions between forest type and peat moisture conditions. *Geoderma* 324, 47–55, https://dio.org/10.1016/j.geoderma.2018.02.029, 2018.

Toczydlowski, A. J. Z., Slesak, R. A., Kolka, R. K., and Venterea, R. T.: Temperature and water-level effects on greenhouse gas fluxes from black ash (Fraxinus nigra) wetland soils in the Upper Great Lakes region, USA. *Appl. Soil Ecol.* 153103565. doi: 10.1016/j.apsoil.2020.103565, 2020.

---

## Author Response (AR1)

**Author's Response**

Author reply in red

We thank both referees for taking the time to read our manuscript "Dynamics of CO2 and CH4 fluxes in Red Sea mangrove soils". The suggestions for improvements from both reviewers have been addressed point-by-point below.

**Reviewer #1**

**This is a very interesting, nicely conducted study that deserves publication. The experiment has been conducted with care and the research question is clear. However, some methods need to be described in more detail, some results should be reported in more detail and some sections of the discussion should be expanded to exploit the scientific benefit that the data harbours.**

We would like to thank reviewer 1 for taking the time to give their thorough feedback and useful comments on our paper. We expanded the different sections as requested to explain the methodology and results in more detail, and also expanded on the discussion. For in depth replies, actions taken, and changes in the manuscript please see the comments below.

**Were there no roots in the soil/sediment cores? If mangrove soils store carbon sequestered by the trees, this carbon must get into the soil somehow.**

Roots and undecomposed organic matter were avoided where possible as the aim was to estimate greenhouse gas (GHG) emissions from the soil rather than the mangrove trees. Given the thickness of roots at the sampling depth, these were relatively easy to avoid. It is well documented that the density of pneumatophores is related to GHG emissions (e.g. Lin et al, 2021; Sheng et al, 2021). The soil organic carbon comes from a combination of decomposed mangrove roots, leaves, and organic matter from other ecosystems (allochthonous inputs), which is decayed by primary consumers. We have clarified this in our methods.

Ln 108-110: During sampling roots and undecomposed organic matter were avoided as the aim was to estimate GHG emission from the soil rather than the mangrove trees.

**Maybe add a sentence elaborating how this aridity influences mangrove growth. Does it also play a role in selecting the dominant mangrove species?**

Thank you for the suggestion. We updated the text to clearly define *Avicennia marina* as the dominant mangrove species in the Red Sea and added more information on the adaptations of A. marina

Ln 42-46: Consequently, *Avicennia marina* is the dominant mangrove species in the Red Sea, existing at the thresholds of its physiological tolerance. It is one of the most highly adapted mangrove species to the high salinity and aridity, and found predominantly as monospecific mangrove stands (Khalil, 2015). *Rhizophora mucronata* are also found within the Red Sea but predominantly in Southern regions where there is lower salinity (Khalil, 2015).

L 48:49: The conditions in the Red Sea result in reduced growth of *A. marina* with trees only reaching 2-3 meters compared to over 16 meters in Australia (Mackey, 1993).

**L 44: Add "also" following "…mangroves is…" as the small tidal range is probably not the only constraint on mangrove distribution.**

L 47: Text has been updated

**L 59: What did Sea et al. (2018) find out? It is only one study, but probably one with some results**

We have updated the text to give a brief insight into the main findings on GHG flux magnitude from Sea et al (2018).

L 63-64: These fluxes ranged from $-3452$ to $7500$ $\mu$mol $CO_2$ m$^{-2}$ d$^{-1}$ and $0.9$ to $13.3$ $\mu$mol $CH_4$ m$^{-2}$ d$^{-1}$ across different locations in the Red Sea (Sea et al., 2018).

**L 115-116: I have a hard time imagining that 12h of equilibration time are sufficient. Did you test whether these 12 h suffice indeed?**

This was substandard word choice and we apologize for the confusion. For clarification, the term "Equilibration" was only intended to mean the time between closing the top lid and the T0 measurement, which has now been replaced by the term 'stabilization' (Garcias-Bonet & Duarte, 2017). In April, May and July (2021) collection was in the early morning. After transportation to the lab the cores were placed into the incubator to keep them at target temperature. One by one, bottom and top lid were exchanged from "collection lids" to air-tight lids used for the duration of the experiment. After sealing/closing the lids, we allowed 1 hour before taking the T0 measurement (following the established protocol of Sea et al., 2018). Due

to variability in the tides, we decided to move sampling time of the soil cores to the afternoon on the day before. This was necessary due to unpredictable tidal conditions and logistical challenges of early morning sampling. The cores were then stored in the incubator at target temperature (with open top lid) approximately 12 hours until closing them and taking T0 measurement. Leaving the cores unsealed and undisturbed for this period was to allow for regular gas exchange following disturbance of the soil caused by collection in the field. Moreover, we wanted to avoid creating anoxic conditions in the sediment and water by closing the core too soon before the experiment. In previous studies using similar methodologies (e.g. soil incubation, gas sampling), soil cores were left for a period between 1 hour to 48 hours before the first gas sample collection (Kristensen et al., 2000; Werner et al., 2014; Christiansen et al., 2015; Perez-Villalona et al., 2015; Sea et al., 2018). If there was water present, extra water was sampled with the sediment cores and also placed in the incubator to keep the temperature stable. In the morning of the start of the experiment, water was exchanged, if present, 1 hour before the collection of the first gas sample (T0) (Kristensen et al., 2000; Sea et al., 2018). The average $CO_2$ concentration (ppm) for T0 across the whole duration of the study was 547 ± 88 ppm, within a normal range for a laboratory during a period of low human activity (Hussin et al., 2017). In April, May and Jul the average was 533ppm, and the T0 measurement was not significantly affected by early morning sample collection, compared to collection the afternoon before the T0 measurement.

The text in the methods has been updated to clearly reflect this and minimize ambiguities.

L 111-124: Initially, in April, May, and July (2021) collection of the soil cores was conducted in the early morning hours allowing for sufficient time to transport and process soil cores with a stabilisation period for 1 hour between sealing the core and taking the T0 gas sample at 7am (following the protocol of Sea et al, 2018). Subsequent sampling events, until the study conclusion, were conducted late afternoon on the day before for logistical reasons, with the cores left unsealed in the incubator under darkness to mirror night-time conditions. Leaving the cores unsealed and undisturbed was to allow for regular gas exchange following disturbance of the soil caused by collection in the field, and to avoid the creation of anoxic conditions in the sediment and water overnight before the start of the experiment. If water was present at the time of sample collection, extra water was sampled with the sediment cores and also placed in the incubator to keep the temperature stable. On the morning of the start of the experiment, water was exchanged, and the air-water interface of the sealed cores was allowed to stabilize for 1-hour before the collection of the T0 gas measurement at 7am (Sea et al., 2018). The 1-hour stabilization was not required for cores without the water phase there was no water to exchange, and no water-soil interface to influence gas exchange dynamics. There were no significant differences in T0 concentrations with or without water.

**L 117-118: What is "light"? Laboratory light intensity? Was there a climate chamber to provide the high light intensity prevailing in the region? Or were the cores outside the lab for the "light" phase?**

The text was modified to specify the light intensity within the laboratory incubator, which was 125 micromoles $m^{-2}$ $sec^{-1}$ of light irradiance according to the manufacturer with the lights set to 100% brightness, reflecting the light intensity in the study region.

L 140-142: For the duration of the light condition, incubator lights were set to 100 % intensity at 125 µmol $m^{-2}$ $sec^{-1}$ irradiance (I-30L, Percival, Geneva Scientific LLC, Fontana Wisconsin, USA).

**L 149-150: How was bulk density determined?**

We apologize for the oversight of adding the equation to the main manuscript. The equation has been added to the methods.

L 171-173: Soil organic carbon ($C_{org}$) and inorganic carbon ($C_{inorg}$) for 0-3 cm soil depth was calculated using bulk density (Howard et al., 2014), and with the following formulas (Eqs. 4-6):

$$Soil\ bulk\ density\ (g\ cm^{-3}) = Oven-dry\ sample\ mass\ (g)\ /\ Sample\ volume\ (m^3)$$

**L 185-194: It would be nice to see, somewhere in this passage, the %Corg, which is a useful parameter in soil science.**

%TOC, with standard error, has been added to the results.

L 212-214: The seaward site had a lower $C_{org}$ concentration, averaging 5.53 mg $C_{org}$ $cm^{-3}$ (0.34 % ± 0.017 %) compared to an average of 9.52 mg $C_{org}$ $cm^{-3}$ (0.72 % ± 0.021 %)

**L 310-311: Why is this sentence underlined?**

Thank you for identifying this error. The issue has been fixed.

**L 338-339: Yes, these differences are not fully resolved. But one reason could be different kinds of transport processes inducing CH4 release (ebullitions vas. Diffusive flux). I suggest elaborating this issue a little.**

We agree with the comment. This study did not differentiate between diffusive flux and ebullition, but it is true that variable transport processes are an explanation for these differences. A few sentences on the kinds of transport processes have been added to the discussion.

L 383-391: The causes for the large differences in GHG flux between sites within the same mangrove stand are not fully resolved, although it is likely that there is microscale variation

due, in part, to different gas transport processes. The release of $CH_4$ from the soil via ebullition has particularly high spatial variability within sampling sites (Baulch, 2011; Chuang et al., 2017). Furthermore, the episodic nature of ebullition events may distort the flux calculation which assumes a linear concentration change over time, as is the case with diffusive flux (Jacotot et al., 2018). The possibility of active ebullition in saline, undisturbed mangrove ecosystems require further investigation, as to-date, no study has found ebullition to be a significant pathway of $CH_4$ release under these conditions (Cotovicz et al., 2024). Considering this small-scale variability, it is important to emphasise the need for comprehensive assessments in individual mangrove ecosystems as GHG flux is highly site-specific.

**L 366-384: This issue should be better exploited, also in the results section. Which soil parameters influence acetoclastic and hydrogenotrophic methanogenesis? When do mangroves grow most strongly in the Red sea? This should be the time of a higher rate of acetoclastic CH4 production. Does d13C efflux change with season?**

To clarify, the measurement of acetoclastic and hydrogenotrophic methanogenesis was not incorporated within our study design, so unfortunately, we cannot present this within our results as this would have required additional measurements. E.g. Inhibition of methanogenesis with 2-bromo-ethane sulphonate, measurement of 14C isotopes (Kotsyurbenko et al., 2004), $\delta$13C of the methyl carbon of acetate (ac-methyl) (Penning et al., 2006), or radiotracer incubations (Weston et al., 2014).

We have, however, elaborated on the results of the 13C isotopic signatures by including some additional statistical analysis. The carbon isotope composition of CO2 and CH4 did not significantly change between seasons (Kruskal-Wallis test: $\delta^{13}C\text{-}CO_2$, p = 0.62; $\delta^{13}C\text{-}CH_4$, p = 0.66). There is also no specific season of growth for A. marina in the Red Sea. There is considerable interannual and geographic variability, which is largely dependent on temperature and humidity (Almahasheer et al., 2016). As there is no defined growth period, unfortunately, isotopic signatures and fluxes cannot be analysed in conjunction with this.

However, to fully address this comment, we have included a Spearman correlation matrix including significance values for all soil parameters, 13C signatures, fluxes, and environmental conditions (uploaded as supplementary material). Data from this analysis has been included in results and to strengthen our discussion of drivers of flux. The 'Data analysis' methods section has been updated accordingly

In brief, from this analysis, there were several soil and environmental variables significantly correlated with stable isotope signatures. These variables were; electrical conductivity, core replicate, and inorganic carbon for the top 3cm of soil. The core replicate also correlated

significantly with a number of soil physicochemical variables. Another notable finding of this analysis showed there was no significant statistical relationship using the Spearman correlation coefficient between $\delta^{13}C$-$CH_4$ and dark CO2 flux, contrary to the random forest model, which suggested it had the highest predictive power. Our interpretation of this, is that there is a complex relationship between $\delta^{13}C$-$CH_4$ and dark CO2 flux but agree with the reviewer on the importance of methanogenesis. We have added the above aspects into the relevant sections to enhance the discussion.

L 185-187: A correlation matrix showing significance between GHG fluxes, isotope signatures, soil properties and environmental variables using Spearman rank correlation coefficient was created with the use of 'Scipy' package (v1.11.1) in Python (v3.11.5).

L 264-266: The $\delta^{13}C$ signature of $CO_2$ and $CH_4$ did not change significantly across seasons. However, significant correlations ($p > 0.05$) were observed between core replicates and inorganic carbon ($C_{inorg}$-3cm) with $\delta^{13}C$-$CO_2$, and well as between electrical conductivity ($EC_{1:5}$) and $\delta^{13}C$-$CH_4$ (Fig. S1).

L 423-425: Notably, there was no statistically significant correlation between $\delta^{13}C$-$CH_4$ and dark CO2 flux, contrary to the random forest model, suggesting this finding may be a result of overfitting from the random forest model or there may be more complex non-linear relationships uncovered by machine learning which are not detected by simple correlation.

L 441-443: However, to better understand the origin and fate of $CH_4$ from mangrove soils, methanogenesis should be studied directly through the determination of $\delta^{13}C$ of the methyl group of acetate (Goevert and Conrad, 2009) or an isotope mass-balance approach (Sánchez-Carrillo et al., 2021), along with an investigation of the soil microbial community.

L 470-474: This is supported by the correlation analysis, where the core replicate had no significant relationships with $CO_2$ or $CH_4$ flux under any conditions. However, there were significant relationships between the core and soil physicochemical properties such as $C_{org}$, TN, and $\delta^{13}C$-$CO_2$ (Fig. S1). This is likely due to microscale differences in the deposition of organic matter, and microbial communities, which is an element of natural variation in response to environmental conditions (Padhy et al., 2020).

**Figs 2, 3: I have the impression that most high CO2 and CH4 fluxes occur between March and May, in both years. Is this mirrored by the d13C signature of the fluxes? Is this the season of highest plant growth?**

There is no consistent month for highest plant growth, although there is generally peak flowering and propagule development in November and January (Almahasheer et al., 2016). The highest growth for A. marina between March and May is unlikely as this is when

temperatures are increasing, there is little to no rain, and tidal inundation becomes less frequent. It is more likely GHG flux correlates with environmental conditions favoring enhanced microbial metabolism. For example, from the correlation matrix, light CH4 flux significantly correlated with temperature (p=0.007) and water content (p=0.009). Dark CH4 flux correlated with water content (p=0.043) and electrical conductivity (p=0.018). CO2 correlated with water volume (light conditions, p=0.008; dark conditions, p=0.032). The 13C signature of the fluxes do not have a significant relationship with the magnitude of the flux, except between light CH4 flux and $\delta^{13}$C-CH$_4$ (p=0.005). The significant correlations with GHG fluxes have been added to the results section 'Drivers of flux variation'. As there is some overlap between this comment and the previous comment these results are already discussed above (e.g. L 470-474).

L 295-298: There were several significant correlations relating to GHG flux with environmental and soil properties. $CO_2$ flux demonstrated a significant correlation with water volume under both conditions (light condition, p=0.008; dark condition, p=0.032) (Fig. S1). Light $CH_4$ flux significantly correlated with temperature (p=0.007) and water content (p=0.009) while dark $CH_4$ flux correlated with water content (p=0.043) and electrical conductivity (p=0.018). (Fig. S1)

L 422-424: Notably, there was no statistically significant correlation between $\delta^{13}$C-CH$_4$ and dark CO2 flux, contrary to the random forest model, suggesting this finding may be a result of overfitting from the random forest model or there may be more complex non-linear relationships uncovered by machine learning which are not detected by simple correlation.

**Figs 5, 6: The predictive power of the year of sampling is interesting. This should be discussed some more.**

The growth and flowering cycles of A. marina mangroves in the Red Sea are not annual (Almahasheer et al., 2016). Potentially, the growth cycle of the studied mangrove stand may have changed across the multi-year duration of the study, thus giving the year a high predictive power in the random forest modelling, but this cannot be verified. Additionally, water was present during 4 of the 5 months sampled in 2021, whereas other years were dominated by dry sampling conditions, this may be artificially inflating the predictive power of the year. There were also climatic variables and extreme weather patterns for the region, particularly in 2023 which may explain the predictive power of the year (Van Dijk et al., 2023).

L 459-468: In both models, the year had the second-highest predictive importance. There are a few theories for the importance of this factor. The growth and flowering cycles of *A. marina* mangroves in the Red Sea are not annual (Almahasheer et al., 2016). In theory, increased growth over a given year may result in increased soil carbon pools for microbial respiration, directly impacting GHG flux. However, this cannot be tested as mangrove growth was not measured in the present study. Alternatively, the importance of the year of sampling may be

artificially inflated in our models due to the presence of water during 4 of the 5 months sampled in 2021 while subsequent years were dominated by dry sampling conditions. However, there were also climatic variables and extreme weather patterns for the region across the 3-year period. 2023 had widespread greening due to higher-than-average rainfall (Van Dijk et al., 2023), potentially also facilitating mangrove growth. It is likely that a combination of these 3 factors explain the predictive importance of the sampling year, and emphasise the importance of long-term flux measurements to capture variations resulting from climatic changes, and perennial life-cycles.

References:

Ahmed, E. A., & Abdel-Hamid, K. A. (2007). Zonation pattern of Avicennia marina and Rhizophora mucronata along the Red Sea Coast, Egypt. World Applied Sciences Journal, 2(4), 283-288.

Almahasheer, H., Duarte, C. M., & Irigoien, X. (2016). Phenology and Growth dynamics of Avicennia marina in the Central Red Sea. Scientific reports, 6(1), 37785.

Baulch, H. M., Dillon, P. J., Maranger, R., & Schiff, S. L. (2011). Diffusive and ebullitive transport of methane and nitrous oxide from streams: Are bubble-mediated fluxes important?. Journal of Geophysical Research: Biogeosciences, 116(G4).

Chuang, P. C., Young, M. B., Dale, A. W., Miller, L. G., Herrera-Silveira, J. A., & Paytan, A. (2017). Methane fluxes from tropical coastal lagoons surrounded by mangroves, Yucatán, Mexico. Journal of Geophysical Research: Biogeosciences, 122(5), 1156-1174.

Cotovicz Jr, L. C., Abril, G., Sanders, C. J., Tait, D. R., Maher, D. T., Sippo, J. Z., ... & Santos, I. R. (2024). Methane oxidation minimizes emissions and offsets to carbon burial in mangroves. Nature Climate Change, 14(3), 275-281.

Christiansen, J. R., Outhwaite, J., & Smukler, S. M. (2015). Comparison of CO2, CH4 and N2O soil-atmosphere exchange measured in static chambers with cavity ring-down spectroscopy and gas chromatography. Agricultural and Forest Meteorology, 211, 48-57.

Goevert, D., & Conrad, R. (2009). Effect of substrate concentration on carbon isotope fractionation during acetoclastic methanogenesis by Methanosarcina barkeri and M. acetivorans and in rice field soil. Applied and Environmental Microbiology, 75(9), 2605-2612.

Howard, J., Hoyt, S., Isensee, K., Telszewski, M., Pidgeon, E. (eds.) (2014). Coastal Blue Carbon: Methods for assessing carbon stocks and emissions factors in mangroves, tidal salt marshes, and seagrasses. Conservation International, Intergovernmental Oceanographic Commission of UNESCO, International Union for Conservation of Nature. Arlington, Virginia, USA.

Hussin, M., Ismail, M. R., & Ahmad, M. S. (2017). Air-conditioned university laboratories: Comparing CO2 measurement for centralized and split-unit systems. Journal of King Saud University-Engineering Sciences, 29(2), 191-201.

Khalil, A. S. (2015). Mangroves of the red sea. The Red Sea: The formation, morphology, oceanography and environment of a young ocean basin, 585-597.

Kotsyurbenko, O. R., Chin, K. J., Glagolev, M. V., Stubner, S., Simankova, M. V., Nozhevnikova, A. N., & Conrad, R. (2004). Acetoclastic and hydrogenotrophic methane production and methanogenic populations in an acidic West-Siberian peat bog. Environmental microbiology, 6(11), 1159-1173.

Kristensen, E., Andersen, F. Ø., Holmboe, N., Holmer, M., & Thongtham, N. (2000). Carbon and nitrogen mineralization in sediments of the Bangrong mangrove area, Phuket, Thailand. Aquatic Microbial Ecology, 22(2), 199-213.

Li, H., Dai, S., Ouyang, Z., Xie, X., Guo, H., Gu, C., ... & Zhao, B. (2018). Multi-scale temporal variation of methane flux and its controls in a subtropical tidal salt marsh in eastern China. Biogeochemistry, 137, 163-179.

Lin, C. W., Kao, Y. C., Lin, W. J., Ho, C. W., & Lin, H. J. (2021). Effects of pneumatophore density on methane emissions in mangroves. Forests, 12(3), 314. Mackey, A. Biomass of the mangrove Avicennia marina (Forsk.) Vierh. near Brisbane, south-eastern Queensland. Marine and Freshwater Research 44, 721–725 (1993).

Penning, H., Claus, P., Casper, P., & Conrad, R. (2006). Carbon isotope fractionation during acetoclastic methanogenesis by Methanosaeta concilii in culture and a lake sediment. Applied and Environmental Microbiology, 72(8), 5648-5652.

Pérez-Villalona, H., Cornwell, J. C., Ortiz-Zayas, J. R., & Cuevas, E. (2015). Sediment denitrification and nutrient fluxes in the San José Lagoon, a tropical lagoon in the highly urbanized San Juan Bay Estuary, Puerto Rico. Estuaries and Coasts, 38, 2259-2278.

Sánchez-Carrillo, S., Garatuza-Payan, J., Sánchez-Andrés, R., Cervantes, F. J., Bartolomé, M. C., Merino-Ibarra, M., & Thalasso, F. (2021). Methane production and oxidation in mangrove soils assessed by stable isotope mass balances. Water, 13(13), 1867.

Sheng, N., Wu, F., Liao, B., & Xin, K. (2021). Methane and carbon dioxide emissions from cultivated and native mangrove species in Dongzhai Harbor, Hainan. Ecological Engineering, 168, 106285.

Van Dijk, A.I.J.M., H.E. Beck, E. Boergens, R.A.M. de Jeu, W.A. Dorigo, T. Frederikse, A. Güntner, J. Haas, J. Hou, W. Preimesberger, J Rahman, P.R. Rozas Larraondo, R. van der Schalie (2024) Global Water Monitor 2023, Summary Report. Global Water Monitor (www.globalwater.online)

Weston, N. B., Neubauer, S. C., Velinsky, D. J., & Vile, M. A. (2014). Net ecosystem carbon exchange and the greenhouse gas balance of tidal marshes along an estuarine salinity gradient. Biogeochemistry, 120, 163-189.

Werner, C., Reiser, K., Dannenmann, M., Hutley, L. B., Jacobeit, J., & Butterbach-Bahl, K. (2014). $N_2O$, NO, $N_2$ and $CO_2$ emissions from tropical savanna and grassland of northern Australia: an incubation experiment with intact soil cores. Biogeosciences, 11(21), 6047-6065

Reviewer #2

**This manuscript investigates CO2 and CH4 fluxes in arid mangroves along the Red Sea. The main findings are GHG fluxes offsets 95% of soil carbon burial in seaward mangrove sites and become net sources during high emission events. However, when total alkalinity enhancement is incorporated, < 4% of carbon sequestration potential is offset by the GHG fluxes. The study also finds that temperature is the most important single variable in predicting CO2 flux under light conditions, second only to the year of sampling due to temporal interannual variability. This study also looked at the relationship between isotopic signature and found a negative correlation between δ13C-CH4, and CO2 flux in both dark and light conditions, which offer insights into how microbial processing are affecting resulting GHG fluxes. Overall, It's a very well written and novel piece of research.**

We thank reviewer 2 for taking the time to review our manuscript and providing valuable comments. We appreciate the positive feedback and have addressed the specific points below.

**The main recommendation I have is for authors to include a couple of sentences to acknowledge limitations related to how incubation technique used in this study could have affected GHG gases relative to the field-based observation such as static chambers and continuous eddy covariance.**

We agree this is an important limitation to consider. We have added a short discussion on limitations of incubation studies and issues with comparison across different methods (e.g. in situ, ex situ). We have justified the reasons for choosing to conduct incubations. Namely, this method gives a better ability to control and manipulate conditions, e.g. constant temperature, and consistent light intensity, which is useful for determining drivers of GHG flux. Although this is an area that could warrant an extensive discussion, we have tried to keep it brief but informative.

L 337-354: While comparisons can, and should, be drawn across different studies, the methodology of the study should be considered when interpreting results. For example, in-situ studies have the advantage of natural conditions with minimal disturbance caused by sampling whereas *ex-situ* studies, such as incubation techniques, allow for greater control of variables but typically cannot entirely replicate *in situ* conditions such as diel temperature variation, changes in light intensity and meteorological conditions (Toczydlowski et al., 2020; Sjögersten et al., 2018). For example, one study found mangrove ecosystem flux of $CH_4$ was the most

variable on a daily basis due to meteorological variables and plant activities, both of which were excluded in this study (Liu et al., 2022). However, this study utilized incubations to maintain stringent control of environmental variables during the measurement period. The caveat of this approach is that it limits applicability to field conditions, but is useful in separating the effects of individual drivers of GHG flux variation from mangrove soil and minimising the number of confounding variables (Bond-Lamberty et al., 2016). An additional element of variation comes from different measurement techniques, as results can differ markedly between laser-based spectrometers, chamber-based systems, and eddy covariance measurements (Brannon et al., 2016; Podgrajsek et al., 2014). All studies compared in Table 3 are of *in situ* design, but there are a range of techniques and calculations used. These elements of variability complicate comparison across studies. There is often a large variation in GHG flux across studies and it should be considered whether this variation is due to environmental conditions or different study designs. For example, in the same study site, $CH_4$ fluxes from eddy covariance measurements have been lower than closed static chamber designs (Gnanamoorthy et al., 2022).

**Tables 1 and 2. Consider adding significance test results to this table, e.g., compact letter display**

This is a very useful suggestion, which neatly adds a substantial additional information to our results. CLD has been added to Tables 1 and 2. The methods have been updated to include the significance tests conducted for the table.

L 184-185: Differences in mean soil properties and GHG flux between sampling sites, and wet and dry conditions were evaluated for significance by means of Mann-Whitney U test in R Studio (v.4.1.2).

**Figure 4. I'd remove the left panel. I didn't find this zoomed in graph helpful to visualize and understand your results.**

This panel was intended to show the differences in the range and median fluxes between the sea-air interface from the landward and seaward site, which is otherwise obscured by the much larger range of fluxes from the soil-air interface from the landward site, although we can see how this may be visually misleading. Therefore, the zoomed in element of Figure 4 has been removed and the single figure has been enlarged to fit the page, making the differences in flux easier to see.

**I agree with you. No one study will ever account for all possible drivers of GHG fluxes. And you are right, lots of these variables can be autocorrelated or have multicollinearity issues. But the**

**relative importance you found could have been very different had you included, say, for example, ammonium or Fe2 in your analyses, right? With that in mind, I think you could offer a sentence or two on this limitation and implications for follow up studies.**

Agreed, our random forest models can only include the variables measured so the results cannot not be taken as absolute variable importance. The text has been updated to more clearly reflect this through adding to the discussion regarding the limited number of soil and environmental properties included in the study and scope for further research on this. We clarified that the random forest models only considered the variables we chose to measure, and are not representative of all soil, temporal and environmental properties.

L 409-417: However, there were variables mentioned above that were found to be important in GHG flux in other studies but were not measured in this study, for example, ammonium, iron, and soil grain size. There are limitations on the number of variables relative to a fairly small number of observations as in this study (Kiers and Smilde, 2007), along with practical limitations of time and resources. There is substantial scope in future research to comprehensively investigate more variables than those reported here over a longer sampling period, or with more frequent observations. An analysis of a greater number of chemical and physical characteristics of the soil beyond carbon and nitrogen would be particularly relevant for GHG flux (Nóbrega et al., 2014; Chen et al., 2010). This limitation must be acknowledged when interpreting our results as there may have been significantly important factors which were not measured and thus not considered in our analysis of the most important drivers of GHG flux.

**Ln 43. 'physiological' or 'ecophysiological' instead?**

L 43: Replaced physiochemical with physiological

**Ln 104. 'cores' instead of 'scores'**

Text has been corrected

**Ln 268. Remove 'good' or replace it by 'high'.**

This sentence has been rephrased

L 315-316: Although the remaining 13 variables all had a feature importance below 0.1 this combination contributed towards an $R^2$ score of 0.63.

**Ln 313. below 'the' salinity or below 'salinities'**

L 380: Changed to 'below the salinity'

**Ln 317. Remove the first 'is' from 'this is method is'**

Text has been updated.

**Ln 317. 'plotsseaaaaaaaaaaaaa'?**

Thank you for identifying this error. The correction has been made

**Ln 320. 'physico-chemical' instead?**

Indeed, 'physiochemical' should read as 'physicochemical'. Physiochemical has been replaced by physicochemical for all occurrences within the manuscript.

References:

Bond-Lamberty, B., Smith, A. P., and Bailey, V.: Temperature and moisture effects on greenhouse gas emissions from deep active-layer boreal soils. Biogeosciences, 13(24), 6669-6681, https://doi.org/10.5194/bg-13-6669-2016, 2016.

Gnanamoorthy, P., Chakraborty, S., Nagarajan, R., Ramasubramanian, R., Selvam, V., Burman, P. K. D., ... and Zhang, Y.: Seasonal variation of methane fluxes in a mangrove ecosystem in south India: An eddy covariance-based approach. Estuaries and Coasts, 1-16, https://doi.org/10.1007/s12237-021-00988-1, 2022.

Kiers, H. A., & Smilde, A. K.: A comparison of various methods for multivariate regression with highly collinear variables. Statistical Methods and Applications, 16, 193- 228, https://doi.org/10.0017/s10260-006-0025-5, 2007.

Liu, J., Valach, A., Baldocchi, D., & Lai, D. Y.: Biophysical controls of ecosystem-scale methane fluxes from a subtropical estuarine mangrove: Multiscale, nonlinearity, asynchrony and causality. Global Biogeochemical Cycles, 36(6), e2021GB007179, https://agupubs.onlinelibrary.wiley.com/doi/full/10.1029/2021GB007179, 2022.

Nóbrega, G. N., Ferreira, T. O., Neto, M. S., Queiroz, H. M., Artur, A. G., Mendonça, E. D. S., ... & Otero, X. L. (2016). Edaphic factors controlling summer (rainy season) greenhouse gas emissions (CO2 and CH4) from semiarid mangrove soils (NE-Brazil). Science of the Total Environment, 542, 685-693, https://doi.org/10.1016/j.scitotenv.2015.10.108, 2014.

Podgrajsek, E., Sahlée, E., Bastviken, D., Ho lst, J., Lindroth, A., Tranvik, L., et al.: Comparison of floating chamber and eddy covariance measurements of lake greenhouse gas fluxes. Biogeosciences 11, 4225 –4233. https://doi.org/10.5194/bg-11-4225-2014, 2014.

Sjögersten, S., Aplin, P., Gauci, V., Peacock, M., Siegenthaler, A., and Turner, B. L.: Temperature response of ex –situ greenhouse gas emissions from tropical peatlands: Interactions between forest type and peat moisture conditions. Geoderma 324, 47–55, https://dio.org/10.1016/j.geoderma.2018.02.029, 2018.

Toczydlowski, A. J. Z., Slesak, R. A., Kolka, R. K., and Venterea, R. T.: Temperature and water-level effects on greenhouse gas fluxes from black ash (Fraxinus nigra) wetland soils in the Upper Great Lakes region, USA. Appl. Soil Ecol. 153103565. doi: 10.1016/j.apsoil.2020.103565, 2020.